# SGD vs GD: Rank Deficiency in Linear Networks

**Aditya Varre**
EPFL
aditya.varre@epfl.ch

**Margarita Sagitova**
EPFL
margarita.sagitova@epfl.ch

**Nicolas Flammarion**
EPFL
nicolas.flammarion@epfl.ch

## Abstract

In this article, we study the behaviour of continuous-time gradient methods on a two-layer linear network with square loss. A dichotomy between SGD and GD is revealed: GD preserves the rank at initialization while (label noise) SGD diminishes the rank regardless of the initialization. We demonstrate this rank deficiency by studying the time evolution of the *determinant* of a matrix of parameters. To further understand this phenomenon, we derive the stochastic differential equation (SDE) governing the eigenvalues of the parameter matrix. This SDE unveils a *replusive force* between the eigenvalues: a key regularization mechanism which induces rank deficiency. Our results are well supported by experiments illustrating the phenomenon beyond linear networks and regression tasks.

## 1 Introduction

Deep neural networks have significantly advanced machine learning in recent decades. A key attribute of these models is their ability, despite being heavily overparameterized, to learn effective representations which generalizes well across different tasks. This capability has sparked substantial interest in understanding how neural networks learn internal representations for specific tasks [Bengio et al., 2013]. Gaining deeper insights into these mechanisms is crucial for enhancing model interpretability and refining training and application methodologies in real-world scenarios.

The success in learning these representations is often attributed to the gradient methods used in training. These methods navigate complex non-convex landscapes, finding solutions that not only minimize the training objective but also yield effective representations. They achieve this generalization while avoiding the spurious features that could potentially arise from the models' large number of parameters. Empirical studies have shown that the stochastic noise in gradient algorithms enhances generalization [Keskar et al., 2017] by favoring solutions with simpler structures that mitigate spurious features [Andriushchenko et al., 2022]. This paper address the overarching question:

*How does stochasticity facilitate the discovery of solutions with simplified structures?*

We explore this question using a simplified model: a single hidden-layer linear network. Despite lacking non-linearity, such networks capture some intricate phenomena of real-world deep networks and have been extensively studied to understand convergence [Arora et al., 2019a, Min et al., 2021], learning dynamics [Saxe et al., 2014], and the implicit bias of optimization algorithms [Gunasekar et al., 2017, Soudry et al., 2018]. Our work builds on this foundation by comparing stochastic algorithms with their deterministic counterparts, focusing on how these differences influence the learning of simpler structures.

38th Conference on Neural Information Processing Systems (NeurIPS 2024).

Specifically, we analyze vector regression on two-layer linear networks trained with both gradient flow and stochastic gradient flow methods. Our contributions include:

- In Section 4, we track the evolution of the determinant of the parameter matrix under gradient flow and stochastic gradient flow. We show that stochastic gradient flow drives the determinant towards zero, effectively removing irrelevant direction(s).

- In Section 5, we derive a stochastic differential equation that describes the behavior of the eigenvalues of the parameter matrix. This analysis reveals a repulsive force between eigenvalues that pushes them apart and a geometric Brownian motion that pulls them toward zero.

- In Section 6, we discuss the generalizability of our approach beyond square loss and various noise models, including discrete step sizes. Finally, we present experimental results in Section 7 that support our theoretical findings.

## 2  Related Work

Our work lies at the convergence of distinct research topics:

**Effect of SGD on generalization.** The relationship between the stochasticity of SGD and its generalization capabilities has been extensively examined [Mandt et al., 2016, Jastrzebski et al., 2018, He et al., 2019, Hoffer et al., 2017, Kleinberg et al., 2018]. Notably, SGD tends to yield models with superior generalization compared to gradient descent [Keskar et al., 2017, Jastrzebski et al., 2018, He et al., 2019]. Various explorations into this phenomenon have been conducted through various approaches: hypothesizing that SGD favors flatter minima linked to better generalization, as opposed to sharp minima associated with poor generalization [Hochreiter and Schmidhuber, 1997, Keskar et al., 2017, Andriushchenko et al., 2023], using a random walk on a random landscape model to understand the impact of stochasticity [Hoffer et al., 2017], proposing that the inherent noise in SGD smooths the loss landscape [Kleinberg et al., 2018], and exploring the implications of dynamical stability [Wu et al., 2018].

**Stochastic dynamics and Label Noise.** Recent literature has explored label noise-driven Gradient Descent as an effective method to probe the beneficial impact of stochasticity on generalization, with two distinct perspectives emerging. Firstly, an asymptotic view on general model parametrization is considered, where Blanc et al. [2020], Damian et al. [2021] suggest that stochastic dynamics preferentially optimize a hidden objective linked to the curvature of the loss. In a related vein, Li et al. [2021] demonstrates appropriate limiting dynamics on the manifold of interpolators through time rescaling. Secondly, specifically for diagonal linear networks, HaoChen et al. [2021], Pillaud-Vivien et al. [2022] observe a similar collapsing effect due to label noise but with a finer characterization of the limiting process. Finally, in the absence of label noise, Pesme et al. [2021], Even et al. [2023] have characterized the outcomes of stochastic GF and GD for diagonal linear networks as the solutions to an implicit regularization problem that results in sparser solutions than without stochasticity. Recently, Ghosh et al. [2023] further exhibit a similar sparser features effect for single-neuron autoencoder. Chen et al. [2023] provides a condition under which an invariant set is attractive for SGD — characterizing the local behavior around these sets. The paper also studies linear networks in a teacher-student setup, however due to structured label-noise [Chen et al., 2023, A2 in p.30], the analysis falls short of capturing the repulsive force in the singular values.

**Linear Networks.** The study of two-layer linear networks has been explored extensively, particularly when optimized using gradient flow on the square loss, across various settings including zero-balance initialization and whitened data Fukumizu [1998], Saxe et al. [2014, 2019], Braun et al. [2022]. Early work by Saxe et al. [2014, 2019] elucidates the temporal changes in the singular values of the predictor, assuming decoupled dynamics and a specific data-dependent weight initialization. This condition is broadened by the analyses of Fukumizu [1998] and Braun et al. [2022], Tarmoun et al. [2021], who apply solutions from a matrix Riccati equation to characterize the weights dynamics under full-rank network initialization. Furthermore, Gidel et al. [2019] extends the existing framework by relaxing the whitened data assumption, conducting a perturbation analysis, and discussing the temporal evolution of the weight matrices' singular values. Additionally, Varre et al. [2024] eliminates the need for zero-balanced and full-rank initializations. Their study provides detailed formulas for weight evolution as a function of the initial scale , also studies a simple version of a stochastic flow without the drift. Wang and Jacot [2023] studied the implicit bias of SGD with $\ell_2$-regularization.

**Matrix valued stochastic process and their eigenvalues.** Stochastic process on the space of symmetric (or Hermitian) matrices and the evolution of their eigenvalues are well studied since Dyson [1962]. These techniques were further developed by Bru [1989, 1991] to study perturbations of principal component analysis and the eigenvalues of Wishart processes. Norris et al. [1986], Graczyk and Małecki [2013] applied SDE-based techniques to study the eigenvalues and eigenvectors of Brownian motion on ellipsoids.

# 3   Linear networks and continuous-time gradient method

**Notation** We use $\langle .,. \rangle$ to denote the inner product, i.e., $\langle u, v \rangle = u^\top v$ for vectors, and $\langle A, B \rangle = \mathrm{Tr}\left(AB^\top\right)$ for matrices. $\mathrm{I}_d$ denotes the identity matrix of dimension $d$ and $0_{p \times k}$ denote the matrix with all zero entries of dimension $p \times k$.

**Vector regression.** We study the vector regression problems with inputs $x_1, \ldots, x_n$ in $(\mathbb{R}^p)^n$ and outputs $y_1, \ldots, y_n$ in $(\mathbb{R}^k)^n$. We consider the minimization of the square loss over a class of parametric models $\mathcal{H} = \{f_\theta(\cdot) : \mathbb{R}^p \to \mathbb{R}^k \mid \theta \in \mathbb{R}^d\}$ specified in the next paragraph. The train loss therefore can be written as $\mathcal{L}\left(\theta\right) = \frac{1}{2n} \sum_{i=1}^n \|y_i - f_\theta(x_i)\|^2$.

**Parameterization with a linear network.** We focus on two-layer linear neural networks of width $l \in \mathbb{N}^*$. The model is described by the parameterization $\theta = (\mathbf{W}_1, \mathbf{W}_2)$, where $\mathbf{W}_1 \in \mathbb{R}^{p \times l}$ and $\mathbf{W}_2 \in \mathbb{R}^{l \times k}$, and the function $f_\theta(x) = \mathbf{W}_2^\top \mathbf{W}_1^\top x$. This model is linear with respect to the input $x$. In terms of expressivity, it is comparable to the linear class of predictors, represented as $f_{\boldsymbol{\beta}}(x) = \boldsymbol{\beta}^\top x$, where $\boldsymbol{\beta}$ equals $\mathbf{W}_1 \mathbf{W}_2$. Throughout our analysis, we denote the equivalent linear predictor of the network as $\boldsymbol{\beta}$. A key aspect of this parametrization is that the prediction function $f_\theta$ is positive homogeneous of degree 2 with respect to $\theta$: specifically, for any $\lambda \in \mathbb{R}$, $f_{\lambda\theta} = \lambda^2 f_\theta$. This property mirrors that of two-layer ReLU networks and significantly influences the loss landscape navigated by the parameters $\theta$. It is important to note that this parameterization introduces some redundancy, a single linear predictor $\boldsymbol{\beta}$ can have multiple representations $\mathbf{W}_1, \mathbf{W}_2$ such that $\mathbf{W}_1 \mathbf{W}_2 = \boldsymbol{\beta}$. Some representations have a rich structure whereas other resemble random features. For example, consider the case of scalar regression ($k = 1$), for a vector $\boldsymbol{\beta}$ there exists rich parameterizations where all the neurons, i.e., columns of $\mathbf{W}_1$ align with $\boldsymbol{\beta}$ and also some lazy structures where $\mathbf{W}_1$ resembles a random matrix [Chizat et al., 2019, Varre et al., 2023].

**Train loss.** By defining $X^\top = [x_1, \ldots, x_n]$ and $Y^\top = [y_1, \ldots, y_n]$, the loss function is given by:

$$\mathcal{L}\left(\mathbf{W}_1, \mathbf{W}_2\right) = \frac{1}{2n} \|X\mathbf{W}_1\mathbf{W}_2 - Y\|^2. \tag{3.1}$$

For simplicity, we adjust for the normalization factor $n$ by rescaling the data to $(X, Y) \leftarrow (X/\sqrt{n}, Y/\sqrt{n})$, thereby implicitly considering it in the loss function without directly mentioning $n$ in the formula. Note that the loss is non-convex in $\mathbf{W}_1, \mathbf{W}_2$.

**Gradient flow.** The dynamics induced in parameter space by running GF on Equation (3.1) is given by

$$d\mathbf{W}_1 = -\nabla_{\mathbf{W}_1}\mathcal{L}\left(\mathbf{W}_1, \mathbf{W}_2\right) dt = X^\top(Y - X\mathbf{W}_1\mathbf{W}_2)\mathbf{W}_2^\top dt, \tag{3.2}$$

$$d\mathbf{W}_2 = -\nabla_{\mathbf{W}_2}\mathcal{L}\left(\mathbf{W}_1, \mathbf{W}_2\right) dt = \mathbf{W}_1^\top X^\top(Y - X\mathbf{W}_1\mathbf{W}_2) dt. \tag{3.3}$$

Introducing the block matrix, $\boldsymbol{\Theta} = \left[\mathbf{W}_1^\top \mid \mathbf{W}_2\right] \in \mathbb{R}^{l \times (p+k)}$ and denoting the residual matrix by $\mathbf{R} = X^\top(Y - X\mathbf{W}_1\mathbf{W}_2)$, the evolution of $\boldsymbol{\Theta}$ can be written as

$$d\boldsymbol{\Theta} = \left[d\mathbf{W}_1^\top \mid d\mathbf{W}_2\right] = \left[\mathbf{W}_2\mathbf{R}^\top dt \mid \mathbf{W}_1^\top \mathbf{R} dt\right] = \left[\mathbf{W}_1^\top \mid \mathbf{W}_2\right] \begin{bmatrix} 0_{p \times p} & \mathbf{R} \\ \mathbf{R}^\top & 0_{k \times k} \end{bmatrix} dt.$$

The gradient flow can therefore be compactly written as

$$d\boldsymbol{\Theta} = \boldsymbol{\Theta}\mathbf{J}dt, \quad \text{where } \mathbf{J} = \begin{bmatrix} 0_{p \times p} & \mathbf{R} \\ \mathbf{R}^\top & 0_{k \times k} \end{bmatrix}. \tag{3.4}$$

The gradient flow (GF), when expressed in this form, reveals an inherent multiplicative structure with respect to $\boldsymbol{\Theta}$ in the gradient of the loss. As we see in subsequent sections, this representation of the gradient flow with block matrices proves to be very convenient.

**Label noise gradient descent.** Label noise gradient descent (LNGD) is a theoretically studied alternative to SGD that mirrors its practical behavior by sharing the geometric properties of the noise Blanc et al. [2020], Damian et al. [2021]. Let $\varepsilon_t \in \mathbb{R}^{n \times k}$, where each entry of $\varepsilon_t$ is an independent Gaussian random variable. At iteration $t$, the labels are perturbed with this Gaussian noise at an intensity $\delta$, i.e., $\widetilde{Y} = Y + \sqrt{\delta}\varepsilon_t$. The LNGD algorithm updates the iterates with a step size $\eta$ in the direction of the gradient computed after the labels have been perturbed, as follows:

$$\mathbf{W}_1^{t+1} = \mathbf{W}_1^t - \eta\nabla_{\mathbf{W}_1}\mathcal{L}\left(\widetilde{Y}, \mathrm{X}, \mathbf{W}_1^t, \mathbf{W}_2^t\right); \quad \mathbf{W}_2^{t+1} = \mathbf{W}_2^t - \eta\nabla_{\mathbf{W}_2}\mathcal{L}\left(\widetilde{Y}, \mathrm{X}, \mathbf{W}_1^t, \mathbf{W}_2^t\right),$$

where, by an abuse of notation, $\mathcal{L}\left(Y, \mathrm{X}, \mathbf{W}_1, \mathbf{W}_2\right) = 1/2\|X\mathbf{W}_1\mathbf{W}_2 - Y\|^2$. The iterates can then be restructured into a block matrix:

$$\mathbf{\Theta}^{t+1} = \mathbf{\Theta}^t - \eta\mathbf{\Theta}^t\mathbf{J}_t - \eta\sqrt{\delta}\mathbf{\Theta}^t\xi_t, \quad \text{where } \xi_t = \begin{bmatrix} 0_{p \times p} & X^\top\varepsilon_t \\ \varepsilon_t^\top X & 0_{k \times k} \end{bmatrix}, \tag{3.5}$$

and $J_t$ is defined as in Equation (3.4).

**Stochastic gradient flow (SGF).** We aim to model the aforementioned LNGD in continuous time using an appropriate SDE. Stochastic continuous-time counterparts of discrete stochastic gradient algorithms are favored for their enhanced amenability to theoretical analysis. We propose the following stochastic differential equation (SDE) to model LNGD in continuous time:

$$\mathrm{d}\mathbf{\Theta} = \mathbf{\Theta}\left[\mathbf{J}\mathrm{d}t + \sqrt{\eta\delta}\mathrm{d}\xi\right], \text{where } \mathrm{d}\xi = \begin{bmatrix} 0_{p \times p} & X^\top\mathrm{d}\mathbf{B}_t \\ \mathrm{d}\mathbf{B}_t^\top X & 0_{k \times k,} \end{bmatrix} \tag{3.6}$$

where $\mathbf{B}_t$ denotes a matrix Brownian motion in $\mathbb{R}^{n \times k}$. LNGD as defined in Equation (3.5), can be interpreted as the the Euler-Maryama discretization of the above SGF with a stepsize $\eta$. Although the inclusion of step size in the continuous-time modeling of an SDE may seem counter-intuitive, it is a necessary component [Li et al., 2019b]. As all the terms of the SDE in Equation (3.6) are polynomial in $\mathbf{\Theta}$, both the drift and diffusion terms are locally Lipschitz continuous. Hence, the solution of the SDE is uniquely defined up to the explosion time $\tau_\infty$ [see, e.g., Khasminskii, 2012]. Furthermore, the explosion time can be proven to be infinite ($\tau_\infty = \infty$ almost surely), by using that the GF does not diverge and applying the techniques outlined by Pillaud-Vivien et al. [2022, Proposition 10].

**Initialization.** The dynamics of gradient methods on homogeneous models are significantly influenced by initialization, which determines the regime they operate in—specifically, the lazy regime for large initializations and the rich regime for small ones [Chizat et al., 2019, Woodworth et al., 2020]. Thus, the scale of initialization has garnered significant interest, particularly its impact on the training of linear and non-linear networks with GD [Woodworth et al., 2020, Boursier et al., 2022]. It is observed that stochastic methods eliminate the dependence on initialization [Pesme et al., 2021].

**Conserved quantities and balanceness.** Gradient flows follow specific conservation laws along their trajectory [Marcotte et al., 2023], maintaining characteristics of the initial conditions. For linear networks, this conservation manifests as the *balanceness property* [Du et al., 2018], described by:

$$\mathbf{\Delta} = \mathbf{W}_1^\top\mathbf{W}_1 - \mathbf{W}_2\mathbf{W}_2^\top = \mathbf{W}_1^\top(0)\mathbf{W}_1(0) - \mathbf{W}_2(0)\mathbf{W}_2^\top(0).$$

As a result, Saxe et al. [2014], Arora et al. [2018, 2019b] have adopted *balanced initialization*, where $\mathbf{\Delta}(0) = 0$, to ensure that weight matrices remain low rank throughout the trajectory. However, unbalanced initialization do not preserve these simple low-rank structures, as aspects of the initial conditions persist.

In contrast, stochastic methods do not adhere to these conservation laws [Ziyin et al., 2023] and the evolution of the imbalance $\mathbf{\Delta}$ for SGF is

$$\mathrm{d}\mathbf{\Delta} = \mathrm{d}\left(\mathbf{W}_1^\top\mathbf{W}_1 - \mathbf{W}_2\mathbf{W}_2^\top\right) = \mathrm{tr}\left(XX^\top\right)\mathbf{W}_2\mathbf{W}_2^\top\mathrm{d}t - k\ \mathbf{W}_1^\top X^\top X\mathbf{W}_1\mathrm{d}t.$$

While there is no diffusion term in the derivative, the matrices remain stochastic and no definitive conclusions can be drawn from this. However, in the case where $k = p$ and $X^\top X = I_p$, it can be shown that $\mathbf{W}_1^\top\mathbf{W}_1 - \mathbf{W}_2\mathbf{W}_2^\top \to 0$, indicating that the stochastic noise eliminates initial imbalance.

**Conclusion.** Understanding how stochastic methods mitigate dependency on initialization requires exploring beyond the evolution of the imbalance $\mathbf{\Delta}$. To this end, we identify and discuss other conserved quantities, such as the determinant of the block matrix $\mathbf{\Theta}^\top\mathbf{\Theta}$ in the following sections.

## 4 Separation between Gradient Flow through determinant

Here, we present our first separation result between GF and SGF. While the determinant of the parameters is preserved in GF, it is driven to zero by the stochasticity of SGF, leading to a simplistic low-rank structure.

### 4.1 Determinant evolution of the gradient flow

The theorem below demonstrates that the determinant of the parameters is preserved in gradient flow.

**Theorem 4.1.** *For the gradient flow defined in Equation* (3.4)*, the following property holds,*

$$\mathrm{d}\big(\det\big(\boldsymbol{\Theta}^\top\boldsymbol{\Theta}\big)\big) = 0.$$

*Hence,* $\det\big(\boldsymbol{\Theta}(t)^\top\boldsymbol{\Theta}(t)\big) = \det\big(\boldsymbol{\Theta}_0^\top\boldsymbol{\Theta}_0\big)$*, where* $\boldsymbol{\Theta}_0 = \boldsymbol{\Theta}(0)$ *is the initialisation at time* $t = 0$*.*

The proof presented in the App. B.1, is based on straightforward computations of the derivative of the determinant and the fact that the matrix $\mathbf{J}$ has zero trace. We note that the simplicity of the proof arises from the strategically chosen block structure of $\boldsymbol{\Theta}$. This result would have been less straightforward with different parametrizations, which likely explains why such a simple finding appears to be novel. The theorem implies that the determinant of $\mathbf{M}$ along the trajectory remains equal to the determinant at initialization. If $\boldsymbol{\Theta}_0^\top\boldsymbol{\Theta}_0$ is full-rank initially, meaning the determinant is non-zero, the theorem ensures that the determinant of $\mathbf{M}$ remains non-zero. Consequently, the rank of $\boldsymbol{\Theta}$ does not diminish along the trajectory. When $l \geq p + k$, i.e., the hidden layer has a large width and $\mathbf{W}_1, \mathbf{W}_2$ are initialized randomly from a Gaussian distribution, $\boldsymbol{\Theta}_0^\top\boldsymbol{\Theta}_0$ has full rank almost surely. The theorem also reveals some implications regarding the impact of initialization scale. Note that $\lambda_{min}(A) \leq \sqrt[n]{\det A}$, indicating that when the scale of initialization is very small, at least one singular value of $\boldsymbol{\Theta}$ is small.

### 4.2 Determinant evolution of the stochastic gradient flow

In contrast, the theorem presented below demonstrates that the determinant of the parameters converges to zero in stochastic gradient flow.

**Theorem 4.2.** *For the SDE, defined in the Equation* (3.6)*, for* $t \leq \tau_\infty$*, the following property holds for the evolution of determinant*

$$\mathrm{d}\big(\det\big(\boldsymbol{\Theta}^\top\boldsymbol{\Theta}\big)\big) = -2\eta\delta k\mathrm{tr}\big(X^\top X\big)\det\big(\boldsymbol{\Theta}^\top\boldsymbol{\Theta}\big)\mathrm{d}t.$$

*Hence,* $\det\big(\boldsymbol{\Theta}(t)^\top\boldsymbol{\Theta}(t)\big) = \det\big(\boldsymbol{\Theta}_0^\top\boldsymbol{\Theta}_0\big)\exp\big\{-2\eta\delta k\mathrm{tr}\big(X^\top X\big)t\big\}$*, where* $\boldsymbol{\Theta}_0$ *is the initialization.*

Although the evolution of the parameters in SGF is random, the evolution of the determinant is deterministic. The theorem highlights a striking phenomenon: the noise in SGF diminishes the determinant along the trajectory, leading to a simplification of the network over time. The larger the noise and the stepsize, the faster the determinant vanishes. The vanishing of the determinant suggests that the rank of the parameters decreases by at least one, effectively eliminating some components. It holds for any initialization of $\boldsymbol{\Theta}_0$ and indicates how the SGF overrides some aspects of initialization. The proof uses the fact that stochastic Brownian term in the SDE, through Itô's calculus, introduces a negative drift, ultimately driving the determinant to zero (refer to B.3 for the proof).

**Limitations.** Given the large width of the hidden layer, the determinant converging to zero does not fully reveal the complexity of the situation. It merely indicates that at least one singular value is approaching zero. Furthermore, the theorem provides limited insights when the determinant is already 0 at initialization, $\det\boldsymbol{\Theta}_0 = 0$ which happens whenever $l < p + k$. Next, we explore the mechanisms behind this low-rank phenomenon, suggesting that the repulsive forces induced by stochasticity drive the spurious singular values to zero as seen in the right plot of Figure 1.

## 5 Mechanism behind the low-rank phenomenon

In this section, we investigate the evolution of singular values under stochastic training to gain deeper insights into the low-rank phenomenon. To simplify the discussion, throughout the section

we consider the case where $k = 1$ and for notational convenience, we let $\mathbf{W}_1 = \mathbf{W}, \mathbf{W}_2 = \mathbf{a}$. Additionally, we assume that $l \leq p$, however the results can be extended to any $l$.

**Warm-up: Comparison with diagonal networks.** Let $\mathbf{W} = \mathbf{U}\Sigma\mathbf{V}^\top$ be the singular value decomposition (assuming $l \leq p$). The predictor $\boldsymbol{\beta}$ can be expressed as

$$\mathbf{Wa} = \mathbf{U}\Sigma\mathbf{V}^\top\mathbf{a} = \mathbf{U}\left[\boldsymbol{\sigma} \odot \mathbf{c}\right], \text{ where } \mathbf{c} = \mathbf{V}^\top\mathbf{a}.$$

This expression reveals a Hadamard product between $\boldsymbol{\sigma}$ and $\mathbf{c}$, reminiscent of diagonal networks which are widely studied to understand the nonconvex dynamics of gradient algorithms [Woodworth et al., 2020, Pesme et al., 2021, Pillaud-Vivien et al., 2022]. In the context of diagonal networks, SGD is known to provably induce sparsity in predictions. Similarly, for linear networks, SGF may induce sparsity in terms of the singular value $\sigma$. We next derive the SDE governing the evolution of the singular values $\Sigma$ of the weight matrix to gain a clearer understanding of the low-rank phenomenon.

**Scalar Regression.** We assume that the data is isotropic, i.e., $X = \mathrm{I}_p$. Under these conditions, the loss function for scalar regression can be written as

$$\mathcal{L}\left(\mathbf{W}, \mathbf{a}\right) = \frac{1}{2}\|y - \mathbf{Wa}\|^2. \tag{5.1}$$

We train the above objective with SGF, formulated as follows,

$$d\mathbf{W} = (y - \mathbf{Wa})\mathbf{a}^\top dt + \sqrt{\eta\delta}\, d\mathbf{B}_t\mathbf{a}^\top; \qquad d\mathbf{a} = \mathbf{W}^\top(y - \mathbf{Wa})dt + \sqrt{\eta\delta}\, \mathbf{W}^\top d\mathbf{B}_t. \tag{5.2}$$

where $\mathbf{B}_t$ is the standard Brownian motion in $\mathbb{R}^p$. For analytical convenience, we rescale the time $t \to {}^t/_{\eta\delta}$ and use the process $d\mathbf{X} = {}^1/_{\eta\delta}(y - \mathbf{Wa})dt + d\mathbf{B}_t$. The SGF can then be rewritten as,

$$d\mathbf{W} = d\mathbf{Xa}^\top; \qquad d\mathbf{a} = \mathbf{W}^\top d\mathbf{X}. \tag{5.3}$$

Our focus is on understanding the evolution of the singular values of the matrix $\mathbf{W}$. This aim is facilitated by considering the symmetric matrix $\mathbf{M} = \mathbf{W}^\top\mathbf{W}$, whose eigenvalues are the squares of the singular values of $\mathbf{W}$. Taking the derivative of $\mathbf{M}$, we find

$$d\mathbf{M} = d\mathbf{W}^\top\mathbf{W} + \mathbf{W}^\top d\mathbf{W} + d\mathbf{W}^\top d\mathbf{W} = \mathbf{a}d\mathbf{X}^\top\mathbf{W} + \mathbf{W}^\top d\mathbf{Xa}^\top + p\mathbf{aa}^\top dt. \tag{5.4}$$

Note that $dxdy$ represents $d[x, y]$ for any continuous semi-martingales $x, y$ [see, e.g., Ikeda and Watanabe, 1981, chapter 3 for reference].

**Eigenvalues of a matrix-valued stochastic process.** We leverage tools from the study of eigenvalues of matrix-valued stochastic processes [Bru, 1989, Graczyk and Małecki, 2013] to derive the evolution of the eigenvalues of $\mathbf{M}$ in the theorem that follows.

**Theorem 5.1.** *Let* $\mathbf{s}_1 > \ldots > \mathbf{s}_l$ *be the order of the eigenvalues of the matrix* $\mathbf{M}$ *defined by Equation* (5.4). *Let the collision time for the eigenvalues be defined as*

$$\tau = \{\inf t : \mathbf{s}_i(t) = \mathbf{s}_j(t) \text{ for } 1 \leq i \neq j \leq l\}. \tag{5.5}$$

*For $t \leq \tau$, the eigenvalues are semi-martingales given by the solution of the following SDE*

$$d(\mathbf{s}_i) = p\mathbf{c}_i^2\, dt + \sum_{\substack{j=1, \\ j \neq i}}^{l} \frac{\mathbf{s}_i\mathbf{c}_j^2 + \mathbf{s}_j\mathbf{c}_i^2}{\mathbf{s}_i - \mathbf{s}_j}dt + 2\sqrt{\mathbf{s}_i\mathbf{c}_i^2}\left(d\tilde{\mathbf{X}}\right)_i \tag{5.6}$$

*where* $\mathbf{c} = \mathbf{V}^\top\mathbf{a}$ *and* $\left(d\tilde{\mathbf{X}}\right)_i = {}^1/_{\eta\delta}\left(\langle\mathbf{u}_i, y\rangle - \sqrt{\mathbf{s}_i\mathbf{c}_i^2}\right)dt + d\varepsilon_i$ *with* $\mathbf{u}_i$ *being the* $i^{th}$ *column of* $\mathbf{U}$ *and* $(\varepsilon_0, \ldots, \varepsilon_{l-1})$ *is the standard Brownian motion in* $\mathbb{R}^l$. *The evolution of* $\mathbf{c}_i$ *and* $\mathbf{U}$ *are presented in the appendix* B.5.

This theorem can be interpreted as the stochastic counterpart to the evolution of eigenvalues previously described for linear networks by Arora et al. [2019c], Varre et al. [2023]. The derivation of the eigenvalues is inspired by the work of Bru [1989].

The evolution of the eigenvalues features a key term highlighted in Equation (5.6) consisting of the sum of skew-symmetric elements ${}^{\mathbf{s}_i\mathbf{c}_j^2 + \mathbf{s}_j\mathbf{c}_i^2}/_{\mathbf{s}_i - \mathbf{s}_j}$. For a pair of indices $(i_0, j_0)$ with $i_0 < j_0$ and thus $\mathbf{s}_{i_0} > \mathbf{s}_{j_0}$, the term ${}^{\mathbf{s}_{i_0}\mathbf{c}_{j_0}^2 + \mathbf{s}_{j_0}\mathbf{c}_{i_0}^2}/_{\mathbf{s}_{i_0} - \mathbf{s}_{j_0}}$ positively influences the evolution of the larger eigenvalue $d\mathbf{s}_{i_0}$ and negatively affects the smaller eigenvalue $d\mathbf{s}_{j_0}$. Therefore, this force is repulsive,

driving the eigenvalues apart and increasing their gap. Another factor influencing the dynamics is the presence of Geometric Brownian motion, where the singular value $\sigma_i$ multiplicatively influences the Brownian motion as $\sqrt{\mathbf{s}_i \mathbf{c}_i^2}\left(\mathrm{d}\tilde{\mathbf{X}}\right)_i$, similar to what is observed in diagonal linear networks (refer to the previous discussion for similarities). This effect tends to pull the singular values toward zero. Together with the fact that $(\mathbf{s}_i, \mathbf{c}_i) = (0, 0)$ represents a fixed point of the dynamics, these two forces collectively push redundant singular values toward zero.

To further understand the interplay of repulsive forces and geometric Brownian motion, we consider the evolution of the smaller singular value $\mathbf{s}_p$ for $l = p$. Using the Ito chain rule, we analyze the evolution of $\log \mathbf{s}_p$, expressed as,

$$\mathrm{d}(\log \mathbf{s}_p) = p\frac{\mathbf{c}_p^2}{\mathbf{s}_p}\,\mathrm{d}t + \frac{1}{\mathbf{s}_p}\sum_{\substack{j=1,\\j\neq p}}^{p}\frac{\mathbf{s}_p\mathbf{c}_j^2 + \mathbf{s}_j\mathbf{c}_p^2}{\mathbf{s}_p - \mathbf{s}_j}\,\mathrm{d}t - 2\frac{\mathbf{c}_p^2}{\mathbf{s}_p} + 2\sqrt{\frac{\mathbf{c}_p^2}{\mathbf{s}_p}}\left(\mathrm{d}\tilde{\mathbf{X}}\right)_p.$$

Using that $\mathbf{s}_p\mathbf{c}_j^2 + \mathbf{s}_j\mathbf{c}_p^2 / \mathbf{s}_p - \mathbf{s}_j < -\mathbf{c}_p^2$, for all indices $j$, the repulsive force accumulates to $-(p-1)(\mathbf{c}_p^2/\mathbf{s}_p)$ and the Ito correction term from the logarithm contributes an additional $-2(\mathbf{c}_p^2/\mathbf{s}_p)$ (the GBM component) thus offsetting the positive drift of $p(\mathbf{c}_p^2/\mathbf{s}_p)$. In the case of $l \neq p$, considering a polynomial $x^\alpha$ with an appropriate $\alpha$ would demonstrate similar behaviour. This discussion outlines the forces at play, yet a complete characterization of the solution of the SDE Equation (5.6) remains missing. Moreover, we have not established that the eigenvalues avoid a.s. collision, i.e., the explosion time $\tau_\infty = \infty$ which is in itself a significant challenge [Bru, 1989, Graczyk and Małecki, 2014].

**A simplified two-vector problem.** To enhance our understanding of the SDE governing the evolution of the eigenvalues detailed in Equation (5.6), we consider the large noise limit. In this scenario, the process described in Equation (5.3) simplifies to a purely noise-driven process without drift:

$$\mathrm{d}\mathbf{W} = \mathrm{d}\mathbf{B}_t\mathbf{a}^\top; \qquad \mathrm{d}\mathbf{a} = \mathbf{W}^\top\mathrm{d}\mathbf{B}_t.$$

This SDE exhibits notable symmetry; allowing for an analysis using a matrix with sub-sampled columns. Let $S$ be any subset of $1, \ldots, l$, with $(\mathbf{w}_i)_{i=1}^{l}$ representing the columns of $\mathbf{W}$. We define $\mathbf{W}_S \in \mathbb{R}^{p \times |S|}$ as the subsampled matrix obtained by selecting columns $\mathbf{w}_i$ where $i \in S$, and similarly, we define a subsampled vector $\mathbf{a}_S$ by selecting the corresponding coordinates. The SDE restricted to the set $S$ is structured as follows:

$$\mathrm{d}\mathbf{W}_S = \mathrm{d}\mathbf{B}_t\mathbf{a}_S^\top; \qquad \mathrm{d}\mathbf{a}_S = \mathbf{W}_S^\top\mathrm{d}\mathbf{B}_t.$$

To demonstrate that the columns of $\mathbf{W}$ align, we leverage the symmetry of the SDE by examining the restricted problem on every pair of rows $S = \{i, j\}$, and proving alignment within this subset. This approach leads us to consider the two vector problem ($l = 2$), where $\mathbf{W} = [\mathbf{w}_1|\mathbf{w}_2]$ and $\mathbf{w}_1, \mathbf{w}_2 \in \mathbb{R}^p, \mathbf{a} \in \mathbb{R}^2$. We describe the behavior of the eigenvalues for this two-vector problem in the theorem below.

**Theorem 5.2.** *In the large noise limit, let $\mathbf{s}_0 > \mathbf{s}_1$ be the eigenvalues of $\mathbf{W}$, the following properties hold, for $t \leq \tau$ defined by $\tau = \{\inf t : \mathbf{s}_0(t) = \mathbf{s}_1(t)\}$,*

> *(a) $\mathbf{s}_0, \mathbf{s}_1$ are greater than zero almost surely,*
>
> *(b) for $\alpha = (p-3)/2$, $\mathbf{s}_0^{-\alpha}$ is a super-martingale while $\mathbf{s}_1^{-\alpha}$ is a sub-martingale.*

This model for $l = 2$ mirrors the dynamics of the Wishart process studied by Bru [1991], motivating the exploration of the evolution of an appropriately chosen exponent of $\mathbf{s}_0, \mathbf{s}_1$. The first part of the theorem arises from the fact that $\mathbf{s}_1^{-\alpha}\mathbf{s}_2^{-\alpha}$ is a local continuous martingale that cannot explode to infinity in finite time. The second part highlights a clear separation between the eigenvalues: one is a sub-martingale that consistently increases in expectation, while the other is a super-martingale that diminishes (note that the eigenvalues are raised to a negative power). This dynamic, coupled with the symmetry argument, suggests that for every pair of columns, there is a component that strengthens the alignment through its increases in expectation. Refer to App. B.6 for the proof.

**Conclusion.** In this section, we derive the SDE of eigenvalues for the matrix of parameters evolving under SGF. This derivation provides deeper insights into the mechanisms contributing to low-rank behavior. Specifically, repulsive forces drive the eigenvalues apart, while the geometric Brownian motion pulls them towards zero. These forces, unique to training with SGF, highlight the regularization effects of stochastic methods compared to gradient flow. However, fully characterizing the solution of this SDE remains a challenging open problem we let as future work.

# 6 Generalization to other settings

In this section, we generalize our results beyond the square loss and the label noise gradient flow. We consider the general framework of a loss function over the weight product $\mathbf{W}_1\mathbf{W}_2$ defined as

$$\mathcal{L}\left(\mathbf{W}_1, \mathbf{W}_2\right) = \widehat{\mathcal{L}}(\mathbf{W}_1\mathbf{W}_2) = \mathbb{E}_{(x,y)\sim\mathcal{D}}\left[\ell(\mathbf{W}_1\mathbf{W}_2; x, y)\right],$$

In this framework, the loss function $\ell$ combines the prediction loss directly with the parametrized model $f_\theta$. This approach applies, for example, to classification problems using linear networks where $\ell$ might represent any classification loss and $f_\theta = \mathbf{W}_1\mathbf{W}_2$. It also directly extends to more complex architectures where $f_\theta = \sigma(\mathbf{W}_1\mathbf{W}_2)$ for an activation function $\sigma$, including settings like a self-attention layer with frozen value vectors. We denote the product by $\boldsymbol{\beta} = \mathbf{W}_1\mathbf{W}_2$ noting it solely controls the loss. We investigate the evolution of the weight matrix determinant for a general loss across various algorithms, from gradient flow to gradient descent, and demonstrate that a similar separation occurs due to stochasticity.

**Warm-up: Gradient flow.** The gradient flow on the loss $\mathcal{L}$ can be written as the following,

$$\mathrm{d}\boldsymbol{\Theta} = \boldsymbol{\Theta}\mathbf{J}\mathrm{d}t, \qquad \text{where } \mathbf{J} = \begin{bmatrix} 0_{p\times p} & -\nabla\widehat{\mathcal{L}}(\boldsymbol{\beta}) \\ -\nabla\widehat{\mathcal{L}}(\boldsymbol{\beta})^\top & 0_{k\times k} \end{bmatrix}. \tag{6.1}$$

Following a similar proof as in Theorem 4.1, we obtain that $\mathrm{d}\left(\det\left(\boldsymbol{\Theta}^\top\boldsymbol{\Theta}\right)\right) = 0$. For separable classification problem, the gradient flow converges to infinity [Soudry et al., 2018, Ji and Telgarsky, 2019], hence, after appropriate rescaling, the layers are aligned, as shown by Ji and Telgarsky [2019]. Next, we contrast this result with the outcomes observed in stochastic and discrete algorithms.

**Continuous modelling of SGD.** We consider the SGD algorithm with a batch size $B$. We denote the mini-batch version of the loss functions $\mathcal{L}$ and $\widehat{\mathcal{L}}$ as $\mathcal{L}_B$ and $\widehat{\mathcal{L}}_B$, respectively. The SGD update with stepsize $\eta$ can be represented with the following block structure,

$$\boldsymbol{\Theta}^{t+1} = \boldsymbol{\Theta}^t - \eta\boldsymbol{\Theta}^t\mathbf{J}^t - \eta\boldsymbol{\Theta}^t\xi^t, \quad \text{where } \xi^t = \begin{bmatrix} 0_{p\times p} & -\left(\nabla\widehat{\mathcal{L}}(\boldsymbol{\beta}) - \nabla\widehat{\mathcal{L}}_B(\boldsymbol{\beta})\right) \\ -\left(\nabla\widehat{\mathcal{L}}(\boldsymbol{\beta}) - \nabla\widehat{\mathcal{L}}_B(\boldsymbol{\beta})\right)^\top & 0_{k\times k} \end{bmatrix}.$$

We denote the SGD noise as $g_t = \left(\nabla\widehat{\mathcal{L}}(\boldsymbol{\beta}) - \nabla\widehat{\mathcal{L}}_B(\boldsymbol{\beta})\right)$ and the noise covariance as $\Sigma_t = \mathbb{E}\left[g^t\left(g^t\right)^\top\right]$ where the expectation is over all the minibatches. Following Li et al. [2019a], the SGD update can be modelled with the following SDE,

$$\mathrm{d}\boldsymbol{\Theta} = -\boldsymbol{\Theta}\mathbf{J}\mathrm{d}t - \sqrt{\eta}\mathrm{d}\xi, \text{where } \mathrm{d}\xi = \begin{bmatrix} 0_{p\times p} & -\Sigma_t^{1/2}\mathrm{d}\mathbf{B}_t \\ -\left(\Sigma_t^{1/2}\mathrm{d}\mathbf{B}_t\right)^\top & 0_{k\times k} \end{bmatrix}. \tag{6.2}$$

The main difference with SGF is that, in overparameterized problems, the noise covariance is time-varying and decreases to zero upon convergence. Using Theorem B.3, the evolution of the determinant of $\mathbf{M} = \boldsymbol{\Theta}^\top\boldsymbol{\Theta}$ is given by $\mathrm{d}(\det(\mathbf{M})) = -\eta\det(\mathbf{M})\mathrm{Tr}\left(\Sigma(t)\right)\mathrm{d}t$ and can be explicitly solved as

$$\mathrm{d}(\det(\mathbf{M})(t)) = \det(\mathbf{M}(0))\exp\{-\eta\int_0^t \mathrm{Tr}\left(\Sigma(s)\right)\mathrm{d}s\}.$$

Hence, the decay in the determinant is governed by the integral $\int_0^\infty \mathrm{Tr}\left(\Sigma(t)\right)\mathrm{d}t$ which is a stochastic quantity. $\mathrm{Tr}\left(\Sigma(t)\right)$ represents the strength of the stochastic noise, which, in over-parameterized regression, is proportional to the loss, i.e., $\mathrm{Tr}\left(\Sigma(t)\right) \propto \mathcal{L}\left(\boldsymbol{\Theta}\right)$ [Pesme et al., 2021]. Therefore, the rate of decay in the determinant depends on $\int_0^\infty \mathcal{L}\left((\boldsymbol{\Theta}(t))\right)\mathrm{d}t$, with slower convergence leading to a simpler model at convergence, as observed in the case of diagonal networks by Pesme et al. [2021]. The result above also holds for *non-separable* classification tasks where the noise of SGD drives the determinant to 0, a scenario not covered by the previous analysis of Ji and Telgarsky [2019].

**Discrete gradient algorithms.** We can extend the previous results to discrete (possibly stochastic) gradient algorithm. Both algorithms can be written as

$$\boldsymbol{\Theta}_{t+1} = \boldsymbol{\Theta}_t\left(\mathrm{I}_{p+k} + \eta\mathbf{J}_t\right),$$

for stepsize $\eta$ and $\mathbf{J}_t$ the possibly stochastic block gradient matrix defined in Equation (6.1). In the context of discrete algorithms, the determinant is controlled by the following lemma (refer to B.4 for the proof).

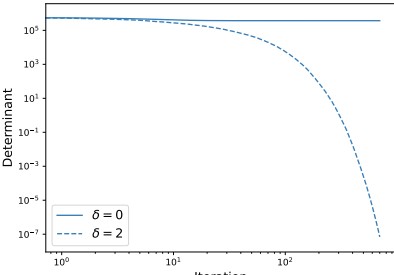 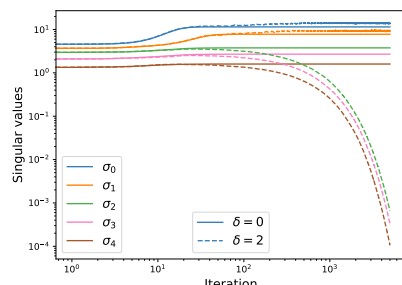

Figure 1: Evolution of the model characteristics for gradient flow ($\delta = 0$) and stochastic gradient flow ($\delta = 2$). Left: Determinant of $\mathbf{M}$. Right: Top-5 singular values of $\mathbf{W}_1$.

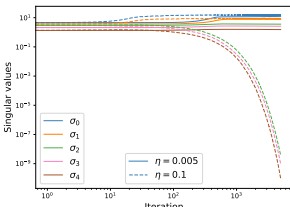 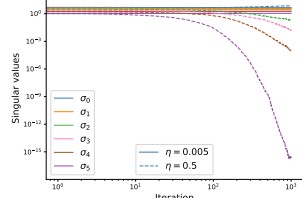 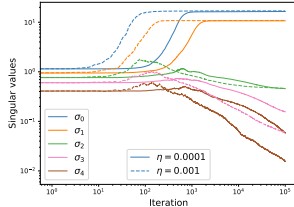

Figure 2: Evolution of the top-5 singular values of $\mathbf{W}_1$ for SGD with small and large stepsizes $\eta$. Left: Regression with MSE loss, linear network. Middle: Classification with logistic loss, linear network. Right: Regression with MSE loss, 2-layer ReLU network.

**Lemma 6.1.** *When $l = p + k$ and $\eta^2 \big\|\mathbf{J}_t\big\|_F^2 \leq 1$, the following property holds for the determinant,*

$$|\det \boldsymbol{\Theta}_{t+1}| \leq \exp\!\left(-\frac{\eta^2}{2}\big\|\mathbf{J}_t\big\|_F^2\right)|\det \boldsymbol{\Theta}_t|.$$

If the factor $\eta^2\big\|\mathbf{J}_t\big\|_F^2 \leq 1$ at every iteration $t$, the determinant is reduced by the discrete step size. However, there is a tradeoff: the sum $S \coloneqq \sum_{t=0}^{\infty} \eta^2\big\|\mathbf{J}_t\big\|_F^2$ can be finite, indicating that it does not completely drive the determinant to zero. Increasing $\eta$ to increase $S$ might lead to instability and divergence. Furthermore, since $\big\|\mathbf{J}_t\big\|_F^2 \propto \mathcal{L}\left(\boldsymbol{\Theta}_t\right)$, there is an additional tradeoff between convergence and the simplicity of the parameters. This illustrates how step sizes that produce non-convergent training loss patterns, such as the catapult effect [Lewkowycz et al., 2020] or the edge of stability mechanisms [Cohen et al., 2020], can simplify the network's parameters.

## 7 Experimental evidence

We consider a regression problem on synthetic data with $n = 1000$ samples of Gaussian data in $\mathbb{R}^5$ ($p = 5$) with labels in $\mathbb{R}^2$ ($k = 2$) generated by some ground truth $\boldsymbol{\beta} \in \mathbb{R}^{5 \times 2}$, the width of the network is $l = 10$. We use Gaussian initialization of the network parameters with entries from $\mathcal{N}(0, 1)$. Experiments details can be found in the appendix C. In the left plot of Figure 1, we show the time evolution of the determinant of matrix $\mathbf{M}$. As suggested by theorems 4.1 and 4.2, in the case without label noise, $\det\left(\boldsymbol{\Theta}^{\top}\boldsymbol{\Theta}\right)$ stays constant, while with the Label Noise of intensity $\delta = 2$ it goes to zero with time. In the right plot of Figure 1, we demonstrate the time evolution of the top-5 singular values of the matrix $\mathbf{W}_1$. Note that in the case of Gradient Flow all except the first $k$ singular values ($\sigma_0$ and $\sigma_1$) stay at the same scale, while adding Label Noise forces smallest $d + l - k$ singular values ($\sigma_2, \sigma_3$, and $\sigma_4$) to tend toward zero. Further experiments illustrate in Figure 2 the evolution of singular values of parameter matrix $\mathbf{W}_1$ when optimized with SGD, for classification tasks and with ReLU network. These results also confirm that the beneficial effects of stochasticity hold in these contexts.

## 8    Conclusion

In this paper, we demonstrate a distinct separation between GF and SGF when trained on linear networks. This separation is obtained by tracking the evolution of the determinant of the parameter matrix. However, while the determinant is a significant factor, it does not fully capture the implicit regularization effects. Notably, the determinant mirrors the imbalance $\mathbf{u}^2 - \mathbf{v}^2$ in diagonal networks represented by $\mathbf{u} \odot \mathbf{v}$, whose dynamics play a crucial role in attuning the implicit regularization across various algorithms [Woodworth et al., 2020, Pesme et al., 2021, Papazov et al., 2024]. Our analysis presents the initial step in deciphering implicit regularization for stochastic methods in linear networks, yet achieving a complete characterization remains a promising direction for future research.

## Acknowledgments and Disclosure of Funding

AV is supported by Swiss data science fellowship. This work was supported by the Swiss National Science Foundation (grant number 212111).

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

# A   Notations

**Notation** $S_d, S_d^+, S_d^{++}$ denote the set of symmetric, positive semi-definite and positive definite matrices in $R^{d \times d}$. We use $\odot$ to denote the Hadamard product.

# B   Proofs

**Theorem B.1.** *For the gradient flow defined in Equation* (3.4)*, the following property holds,*

$$d\big(\det\big(\boldsymbol{\Theta}^\top \boldsymbol{\Theta}\big)\big) = 0.$$

*Hence,* $\det\big(\boldsymbol{\Theta}(t)^\top \boldsymbol{\Theta}(t)\big) = \det\big(\boldsymbol{\Theta}_0^\top \boldsymbol{\Theta}_0\big)$*, where* $\boldsymbol{\Theta}_0 = \boldsymbol{\Theta}(0)$ *is the initialisation at time* $t = 0$*.*

First, we present a proof of this theorem, based on straightforward computations of the derivative of the determinant and the fact that the matrix $\mathbf{J}$ has zero trace.

*Proof.* Let $\mathbf{M} = \boldsymbol{\Theta}^\top \boldsymbol{\Theta}$. The dynamics of $\mathbf{M}$ are governed by the ODE,

$$d\mathbf{M} = d\boldsymbol{\Theta}^\top \boldsymbol{\Theta} + \boldsymbol{\Theta}^\top d\boldsymbol{\Theta} = \boldsymbol{\Theta}^\top \boldsymbol{\Theta} \mathbf{J} dt + \mathbf{J} \boldsymbol{\Theta}^\top \boldsymbol{\Theta} dt = (\mathbf{MJ} + \mathbf{JM}) dt.$$

Using the gradient of the determinant given in Proposition B.2, the determinant of $\mathbf{M}$ evolves as follows,

$$d(\det(\mathbf{M})) = \langle \nabla \det(\mathbf{M}), d\mathbf{M} \rangle = \det(\mathbf{M}) \langle \mathbf{M}^{-1}, \mathbf{MJ} + \mathbf{JM} \rangle dt,$$
$$= \det(\mathbf{M}) \langle \mathbf{M}^{-1}, \mathbf{MJ} \rangle + \langle \mathbf{M}^{-1}, \mathbf{JM} \rangle = 2\det(\mathbf{M}) \langle \mathbf{I}_{p+k}, \mathbf{J} \rangle = 2\det(\mathbf{M}) \mathrm{Tr}(\mathbf{J}).$$

Given that $\mathrm{Tr}(\mathbf{J}) = 0$, it follows that $d(\det(\mathbf{M})) = 0$. $\qquad\square$

**Proposition B.2.** *For any matrix M in* $S_d^{++}$*, the first two derivatives of the determinant of M, denoted by* $\det(M)$ *are the following*

   *(i)* $\nabla \det(M) = \det(M) M^{-1}$

   *(ii) For* $1 \leq a, b, k, l \leq d$*, the second order partial derivative is given by*

$$\frac{\partial^2 \det(M)}{\partial M_{ab} \partial M_{kl}} = \det(M) \left[ (M^{-1})_{ba} (M^{-1})_{lk} - (M^{-1})_{bk} (M^{-1})_{la} \right] \tag{B.1}$$

**Theorem B.3.** *For a stochastic process given by the SDE,*

$$d\boldsymbol{\Theta} = \boldsymbol{\Theta} \left[ \mathbf{J} dt + d\xi \right] \tag{B.2}$$

*with* $\mathrm{Tr}\, \mathbf{J} = \mathrm{Tr}\, \xi = 0$*, the determinant of the* $\mathbf{M} = \boldsymbol{\Theta}^\top \boldsymbol{\Theta}$ *evolves as*

$$d(\det(\mathbf{M})) = -\det(\mathbf{M}) \mathrm{Tr}\left[ d\xi d\xi \right]. \tag{B.3}$$

*Proof.* First, we compute the evolution of $\mathbf{M} = \boldsymbol{\Theta}^\top \boldsymbol{\Theta}$ using the Ito's product rule,

$$d\mathbf{M} = d\big(\boldsymbol{\Theta}^\top \boldsymbol{\Theta}\big) = d\boldsymbol{\Theta}^\top \boldsymbol{\Theta} + \boldsymbol{\Theta}^\top d\boldsymbol{\Theta} + d\boldsymbol{\Theta}^\top d\boldsymbol{\Theta}$$

The last term is interpreted as a derivative of the finite variation and it should be computed using $dt$. $(d\mathbf{B}_t)_{ij} = 0$ and $(d\mathbf{B}_t)_{ij} \cdot (d\mathbf{B}_t)_{kl} = \delta_{i=k \wedge j=l} dt$. Using Eq. (3.6),

$$d\mathbf{M} = [\mathbf{J} dt + d\xi] \boldsymbol{\Theta}^\top \boldsymbol{\Theta} + \boldsymbol{\Theta}^\top \boldsymbol{\Theta} [\mathbf{J} dt + d\xi] + d\xi \boldsymbol{\Theta}^\top \boldsymbol{\Theta} d\xi,$$
$$= \mathbf{JM} dt + \mathbf{MJ} dt + d\xi \mathbf{M} d\xi + d\xi \mathbf{M} + \mathbf{M} d\xi.$$

Using the Ito chain rule, we can compute the evolution of determinant as following,

$$d(\det(\mathbf{M})) = \langle \nabla \det(\mathbf{M}), d\mathbf{M} \rangle + \frac{1}{2} \sum_{a,b,k,l} \frac{\partial^2 \det(\mathbf{M})}{\partial \mathbf{M}_{ab} \partial \mathbf{M}_{kl}} d\mathbf{M}_{ab} d\mathbf{M}_{kl},$$

The first term is

$$\langle \nabla \det(\mathbf{M}), d\mathbf{M} \rangle = \det(\mathbf{M}) \left\langle \mathbf{M}^{-1}, \mathbf{JM}dt + \mathbf{MJ}dt + d\xi\mathbf{M}d\xi + d\xi\mathbf{M} + \mathbf{M}d\xi \right\rangle,$$
$$= 2det(\mathbf{M}) \langle \mathbf{I}_{p+k}, \mathbf{J} \rangle dt + 2\det(\mathbf{M}) \langle \mathbf{I}_{p+k}, d\xi \rangle + \det(\mathbf{M}) \left\langle \mathbf{M}^{-1}, d\xi\mathbf{M}\xi \right\rangle$$

Using the property that $\text{Tr}(\mathbf{J}) = \text{Tr}(d\xi) = 0$. We get that $\langle \nabla \det(\mathbf{M}), d\mathbf{M} \rangle = \left\langle \mathbf{M}^{-1}, d\xi\mathbf{M}\xi \right\rangle$. For the second term

$$\frac{1}{2} \sum_{a,b,k,l} \frac{\partial^2 \det(\mathbf{M})}{\partial \mathbf{M}_{ab} \partial \mathbf{M}_{kl}} d\mathbf{M}_{ab} d\mathbf{M}_{kl} = \frac{1}{2} \sum_{a,b,k,l} \det\mathbf{M} \left[ (\mathbf{M}^{-1})_{ba}(\mathbf{M}^{-1})_{lk} - (\mathbf{M}^{-1})_{bk}(\mathbf{M}^{-1})_{la} \right] d\mathbf{M}_{ab} d\mathbf{M}_{kl},$$

$$= \frac{det(\mathbf{M})}{2} \sum_{a,b,k,l} \left[ (\mathbf{M}^{-1})_{ba}(\mathbf{M}^{-1})_{lk} \right] d\mathbf{M}_{ab} d\mathbf{M}_{kl}$$

$$- \sum_{a,b,k,l} \left[ (\mathbf{M}^{-1})_{bk}(\mathbf{M}^{-1})_{la} \right] d\mathbf{M}_{ab} d\mathbf{M}_{kl},$$

Rearranging the terms in the summation, we get,

$$\sum_{a,b,k,l} \left[ (\mathbf{M}^{-1})_{ba}(\mathbf{M}^{-1})_{lk} \right] d\mathbf{M}_{ab} d\mathbf{M}_{kl} = \sum_{a,b,k,l} \left[ (\mathbf{M}^{-1})_{ba} d\mathbf{M}_{ab} \right] \left[ (\mathbf{M}^{-1})_{lk} d\mathbf{M}_{kl} \right],$$

$$= \sum_{b,l} \left[ \sum_a (\mathbf{M}^{-1})_{ba} d\mathbf{M}_{ab} \right] \left[ \sum_k (\mathbf{M}^{-1})_{lk} d\mathbf{M}_{kl} \right],$$

$$= \sum_{b,l} \left( \mathbf{M}^{-1} d\mathbf{M} \right)_{bb} \left( \mathbf{M}^{-1} d\mathbf{M} \right)_{ll} = \sum_b \left( \mathbf{M}^{-1} d\mathbf{M} \right)_{bb} \sum_l \left( \mathbf{M}^{-1} d\mathbf{M} \right)_{ll},$$

$$= \text{Tr} \left( \mathbf{M}^{-1} d\mathbf{M} \right) \text{Tr} \left( \mathbf{M}^{-1} d\mathbf{M} \right).$$

Similarly for the other term, we get,

$$\sum_{a,b,k,l} \left[ (\mathbf{M}^{-1})_{bk}(\mathbf{M}^{-1})_{la} \right] d\mathbf{M}_{ab} d\mathbf{M}_{kl} = \sum_{a,b,k,l} \left[ (\mathbf{M}^{-1})_{bk} d\mathbf{M}_{kl} \right] \left[ (\mathbf{M}^{-1})_{la} d\mathbf{M}_{ab} \right],$$

$$= \sum_{b,l} \left[ \sum_a (\mathbf{M}^{-1})_{ba} d\mathbf{M}_{al} \right] \left[ \sum_k (\mathbf{M}^{-1})_{bk} d\mathbf{M}_{kl} \right],$$

$$= \sum_b \left[ \sum_l \left( \mathbf{M}^{-1} d\mathbf{M} \right)_{bl} \left( \mathbf{M}^{-1} d\mathbf{M} \right)_{lb} \right] = \sum_b \left( \mathbf{M}^{-1} d\mathbf{M} \mathbf{M}^{-1} d\mathbf{M} \right)_{bb},$$

$$= \text{Tr} \left[ \left( \mathbf{M}^{-1} d\mathbf{M} \right) \left( \mathbf{M}^{-1} d\mathbf{M} \right) \right].$$

Note that the diffusion part of $\mathbf{M}^{-1} d\mathbf{M}$ is $d\xi + \mathbf{M}^{-1} d\xi\mathbf{M}$. Using this

$$\text{Tr} \left( \mathbf{M}^{-1} d\mathbf{M} \right) \text{Tr} \left( \mathbf{M}^{-1} d\mathbf{M} \right) = \text{Tr} \left[ d\xi + \mathbf{M}^{-1} d\xi\mathbf{M} \right] \text{Tr} \left[ d\xi + \mathbf{M}^{-1} d\xi\mathbf{M} \right] = 0,$$

as $\text{Tr} \, d\xi = 0$. For the other term,

$$\text{Tr} \left[ \left( \mathbf{M}^{-1} d\mathbf{M} \right) \left( \mathbf{M}^{-1} d\mathbf{M} \right) \right] = \text{Tr} \left[ \left( d\xi + \mathbf{M}^{-1} d\xi\mathbf{M} \right) \left( d\xi + \mathbf{M}^{-1} d\xi\mathbf{M} \right) \right],$$

$$= 2\text{Tr} \left[ d\xi d\xi \right] + 2\text{Tr} \left[ \mathbf{M}^{-1} d\xi\mathbf{M}d\xi \right].$$

Putting everything together, we get,

$$\frac{1}{2} \sum_{a,b,k,l} \frac{\partial^2 \det(\mathbf{M})}{\partial \mathbf{M}_{ab} \partial \mathbf{M}_{kl}} = -\det\mathbf{M} \left( \text{Tr} \left[ d\xi d\xi \right] + \text{Tr} \left[ \mathbf{M}^{-1} d\xi\mathbf{M}d\xi \right] \right)$$

which gives us

$$d(\det(\mathbf{M})) = -\det(\mathbf{M}) \, \text{Tr} \left[ d\xi d\xi \right].$$

$\square$

**Lemma B.4.** *When $l = p + k$ and $\eta^2 \|\mathbf{J}_t\|_F^2 \leq 1$, the following property holds for the determinant,*

$$|\det \mathbf{\Theta}_{t+1}| \leq \exp\left(-\frac{\eta^2}{2}\|\mathbf{J}_t\|_F^2\right)|\det \mathbf{\Theta}_t|.$$

*Proof.* Note that because of the block structure of the matrix $\mathbf{J}_t$, its nonzero eigenvalues come in $\pm$-pairs: $\pm\sigma_1, \ldots, \pm\sigma_m$, moreover, since $\mathbf{J}_t$ is symmetric, singular values of $\mathbf{J}_t$ are the absolute values of eigenvalues, i.e. $\sigma_1, \ldots, \sigma_m$. Then, the determinant of $\mathbf{\Theta}_{t+1}$ can be written as the following,

$$\det \mathbf{\Theta}_{t+1} = \det \mathbf{\Theta}_t \det (\mathrm{I}_{p+k} + \eta\mathbf{J}_t) = \det \mathbf{\Theta}_t \prod_{i=1}^{m}(1 - \eta^2\sigma_i^2).$$

Using that $1 - x^2 \leq e^{-x^2}$ for all $x$, we can estimate

$$\prod_{i=1}^{m}(1 - \eta^2\sigma_i^2) \leq \exp\left(-\eta^2\sum_{i=1}^{m}\sigma_i^2\right) = \exp\left(-\frac{\eta^2}{2}\|\mathbf{J}_t\|_F^2\right).$$

We obtain the required inequality by observing that $\prod_{i=1}^{m}(1 - \eta^2\sigma_i^2) = \left|\prod_{i=1}^{m}(1 - \eta^2\sigma_i^2)\right|$ since each term $1 - \eta^2\sigma_i^2 \geq 0$ when $\eta^2\|\mathbf{J}_t\|_F^2 < 1$. $\qquad\square$

**Theorem B.5.** *Let $\mathbf{s}_1 > \ldots \mathbf{s}_l$ be the order of the eigenvalues of the matrix $\mathbf{M}$ defined by Equation (5.4). Let the collision time for the eigenvalues be defined as*

$$\tau = \{\inf t : \mathbf{s}_i(t) = \mathbf{s}_j(t) \text{ for } 1 \leq i \neq j \leq l\}. \tag{B.4}$$

*For $t \leq \tau$, the eigenvalues are semi-martingales given by the solution of the following SDE*

$$\mathrm{d}(\mathbf{s}_i) = p\mathbf{c}_i^2 \, \mathrm{d}t + \sum_{\substack{j=1, \\ j \neq i}}^{l} \frac{\mathbf{s}_i\mathbf{c}_j^2 + \mathbf{s}_j\mathbf{c}_i^2}{\mathbf{s}_i - \mathbf{s}_j} \mathrm{d}t + 2\sqrt{\mathbf{s}_i\mathbf{c}_i^2}\left(\mathrm{d}\tilde{\mathbf{X}}\right)_i \tag{B.5}$$

*where $\left(\mathrm{d}\tilde{\mathbf{X}}\right)_i = {}^{1}/_{\eta\delta}\left(\langle\mathbf{u}_i, y\rangle - \sqrt{\mathbf{s}_i\mathbf{c}_i^2}\right)\mathrm{d}t + \mathrm{d}\varepsilon_i$ with $\mathbf{u}_i$ being the $i^{th}$ column of $\mathbf{U}$ and $(\varepsilon_0, \ldots, \varepsilon_{l-1})$ is the standard Brownian motion in $\mathbb{R}^l$. The evolution of $\mathbf{c}_i$ and $\mathbf{U}$ are presented in the appendix.*

*Proof.* The proof follows the approach of Bru [1989]. Let $\mathbf{W} = \mathbf{U}\mathbf{\Sigma}\mathbf{V}^\top$ be the singularvalue decomposition (see Def.D.1 involved with $r = l$ and $l < p$ and it will be the rank). Our focus is on understanding the evolution of the singular values and singular vectors of the matrix $\mathbf{W}$. To derive the evolution of $\mathbf{\Sigma}, \mathbf{V}$ we can consider the eigenvalues and eigenvectors of the PSD matrix process $\mathbf{M}$. Note that $\mathbf{M} = \mathbf{V}\mathbf{\Sigma}^2\mathbf{V}^\top$, let $\mathbf{D} = \mathbf{\Sigma}^2$.

**Evolution of $\mathbf{D}$ and $\mathbf{V}$** Taking the derivative of $\mathbf{M}$, we find

$$\mathrm{d}\mathbf{M} = \mathrm{d}\mathbf{W}^\top\mathbf{W} + \mathbf{W}^\top\mathrm{d}\mathbf{W} + \mathrm{d}\mathbf{W}^\top\mathrm{d}\mathbf{W} = a\mathrm{d}\mathbf{X}^\top\mathbf{W} + \mathbf{W}^\top\mathrm{d}\mathbf{X}a^\top + p\mathbf{a}\mathbf{a}^\top \mathrm{d}t. \tag{B.6}$$

We invoke the theorem D.2 we derived to give the eigenvalues of any matrix valued stochastic process. Note that $\mathbf{V}\mathbf{V}^\top = \mathrm{I}_l$, so some terms of the computation are not required.

$$\mathrm{d}\mathbf{D} = \mathrm{I} \odot \widetilde{\mathbf{N}} \, \mathrm{d}t + \mathrm{I} \odot \mathrm{d}\widetilde{\mathbf{M}} \, \mathrm{d}t + \mathrm{I} \odot \left(\mathrm{d}\widetilde{\mathbf{M}}\left(\mathbf{S} \odot \mathrm{d}\widetilde{\mathbf{M}}\right)\right).$$

and the evolution of the eigenvectors,

$$\mathrm{d}\mathbf{V} = \mathbf{V}\left(\mathbf{Q}_\| \, \mathrm{d}t + \mathbf{S} \odot (\widetilde{\mathbf{N}}\mathrm{d}t + \mathrm{d}\widetilde{\mathbf{M}})\right)$$

where you define,

$$\mathbf{Q}_\| = \frac{\mathrm{I} \odot \left[\left(\mathbf{S} \odot \mathrm{d}\widetilde{\mathbf{M}}\right)\left(\mathbf{S} \odot \mathrm{d}\widetilde{\mathbf{M}}\right)\right]}{2} - \mathbf{S} \odot \left[\left(\mathbf{S} \odot \mathrm{d}\widetilde{\mathbf{M}}\right)\left[\mathrm{d}\widetilde{\mathbf{M}} \odot \mathrm{I}\right]\right] + \mathbf{S} \odot \left(\mathrm{d}\widetilde{\mathbf{M}}\left(\mathbf{S} \odot \mathrm{d}\widetilde{\mathbf{M}}\right)\right)$$

where the matrix $\mathbf{S}$ is given by

$$\mathbf{S}_{ij} = \begin{cases} 0 & \text{if } i = j, \\ (\mathbf{s}_j - \mathbf{s}_i)^{-1} & \text{o.w.} \end{cases}$$

$$\widetilde{\mathbf{N}} = \mathbf{V}^\top (p\mathbf{a}\mathbf{a}^\top)\mathbf{V} = p\mathbf{c}\mathbf{c}^\top.$$

$$\widetilde{d\mathbf{M}} = \mathbf{V}^\top \left[ \mathrm{ad}\mathbf{X}^\top \mathbf{W} + \mathbf{W}^\top d\mathbf{X}\mathbf{a}^\top \right] \mathbf{V},$$

$$= \mathbf{c}d\mathbf{X}^\top \mathbf{U}\mathbf{\Sigma} + \mathbf{\Sigma}\mathbf{U}^\top d\mathbf{X}\mathbf{c}^\top.$$

Note that $\mathbf{\Sigma} = \mathrm{diag}\left((\boldsymbol{\sigma}_0, \ldots, \boldsymbol{\sigma}_{l-1})\right)$ where $\boldsymbol{\sigma}_0 > \boldsymbol{\sigma}_1 \ldots > \boldsymbol{\sigma}_{l-1}$. Let $\mathbf{D} = \mathbf{\Sigma}^2$ and denote the entires of $\mathbf{D}$ as following, $\mathbf{D} = \mathrm{diag}\left((\mathbf{s}_0, \ldots, \mathbf{s}_{p-1})\right)$. Note that

$$\mathbf{U}^\top d\mathbf{X} = \mathbf{U}^\top (\frac{1}{\eta\delta}(y - \mathbf{W}\mathbf{a})dt + d\mathbf{B}_t),$$

$$= \frac{1}{\eta\delta} \left[ \mathbf{U}^\top y - \mathbf{\Sigma}\mathbf{c} \right] dt + \mathbf{U}^\top d\mathbf{B}_t.$$

Using Levy's characterization $\mathbf{U}^\top d\mathbf{B}_t$ is a Brownian motion in $\mathbb{R}^l$, lets call that $d\tilde{\mathbf{B}}_t$. The diffusion part of $\widetilde{d\mathbf{M}}$ (say $d\mathbf{F}$)

$$d\mathbf{F} = \mathbf{\Sigma}\mathbf{V}^\top d\mathbf{B}_t\mathbf{c}^\top + \mathbf{c}d\mathbf{B}_t^\top \mathbf{V}\mathbf{\Sigma},$$

$$= \left(\boldsymbol{\sigma} \odot d\tilde{\mathbf{B}}_t\right)\mathbf{c}^\top + \mathbf{c}\left(\boldsymbol{\sigma} \odot d\tilde{\mathbf{B}}_t\right)^\top$$

$$= d\mathbf{m}_t\mathbf{c}^\top + \mathbf{c}d\mathbf{m}_t^\top$$

where $d\mathbf{m}_t \overset{\text{def}}{=} (\boldsymbol{\sigma} \odot d\tilde{\mathbf{B}}_t)$. We are required to compute $d\mathbf{F}(\mathbf{S} \odot d\mathbf{F})$ to compute the evolution of eigenvalues. Using the lemma D.4, we get

$$d\mathbf{F}(\mathbf{S} \odot d\mathbf{F}) = \mathbf{c}\mathbf{s}^\top \mathbf{S}\mathrm{diag}\left(\mathbf{c}\right)dt - \mathbf{D}\mathrm{diag}\left(\mathbf{S}\mathrm{diag}\left(\mathbf{c}\right)\mathbf{c}\right)dt + \mathbf{D}\mathrm{diag}\left(\mathbf{c}\right)\mathbf{S}\mathrm{diag}\left(\mathbf{c}\right)dt,$$

$$\mathrm{I} \odot [d\mathbf{F}(\mathbf{S} \odot d\mathbf{F})] = \mathrm{I} \odot \left[\mathbf{c}\mathbf{s}^\top \mathbf{S}\mathrm{diag}\left(\mathbf{c}\right)dt - \mathbf{D}\mathrm{diag}\left(\mathbf{S}\mathrm{diag}\left(\mathbf{c}\right)\mathbf{c}\right)dt\right]$$

The element wise computation of this term gives the required result for evolution of eigenvalues.

**Evolution of c.** Note that $c = \mathbf{V}^\top \mathbf{a}$. Computing the derivative using the Ito's product rule, we get,

$$d\mathbf{V}^\top \mathbf{a} = \mathbf{V}^\top d\mathbf{a} + d\mathbf{V}^\top \mathbf{a} + d\mathbf{V}^\top d\mathbf{a},$$

$$= \mathbf{V}^\top d\mathbf{a} + d\mathbf{V}^\top \mathbf{V}\mathbf{V}^\top \mathbf{a} + d\mathbf{V}^\top \mathbf{V}\mathbf{V}^\top d\mathbf{a},$$

$$d\mathbf{V}^\top \mathbf{V} = \left[\left(\mathbf{Q}_\parallel^\top dt - \mathbf{S} \odot d\mathbf{X}\right)\right],$$

$$\mathbf{V}^\top d\mathbf{a} = \mathbf{V}^\top \mathbf{W}^\top d\mathbf{B}_t + \frac{1}{\eta\delta}\left[\mathbf{U}^\top y - \mathbf{\Sigma}\mathbf{c}\right]dt = \mathbf{\Sigma}d\tilde{\mathbf{B}}_t = d\mathbf{m}_t + \frac{1}{\eta\delta}\left[\mathbf{U}^\top y - \mathbf{\Sigma}\mathbf{c}\right]dt,$$

$$d\mathbf{V}^\top \mathbf{V}\mathbf{V}^\top d\mathbf{a} = -(\mathbf{S} \odot d\mathbf{F})d\mathbf{m}_t.$$

$$d\mathbf{V}^\top \mathbf{V}\mathbf{V}^\top d\mathbf{a} = \left[\left(\mathbf{Q}_\parallel^\top dt - \mathbf{S} \odot \left(\widetilde{\mathbf{N}}dt + \widetilde{d\mathbf{M}}\right)\right)\right]\mathbf{c}$$

Using the lemma D.6, D.5, D.4 and computing the element wise summation, we get the following evolution for $d\mathbf{c}$

$$d\mathbf{c}_i = -\frac{1}{2}\sum_{j=1}^{l} \mathbf{S}_{ij}(\mathbf{s}_i\mathbf{c}_j^2 + \mathbf{s}_j\mathbf{c}_i^2)dt - \mathbf{c}_i \sum_{j=1}^{l}(\mathbf{S}_{ij}\mathbf{c}_j^2)\left(\sum_{k \neq i,j} \mathbf{s}_k\mathbf{S}_{ki}\right)$$

$$- (p-2)\mathbf{c}_i \sum_{j=1}^{l} \mathbf{S}_{ij}\mathbf{c}_i^2 dt - \sum_{j=1}^{l} \mathbf{S}_{ij}\mathbf{s}_j dt,$$

$$+ \boldsymbol{\sigma}_i(\mathbf{U}^\top d\mathbf{X})_i(1 - \sum_{j=1}^{l} \mathbf{S}_{ij}\mathbf{c}_j^2) - \mathbf{c}_i \sum_{j} \mathbf{S}_{ij}\boldsymbol{\sigma}_j\mathbf{c}_j(\mathbf{U}^\top d\mathbf{X})_j$$

**Evolution of U.** To compute the evolution of $\mathbf{U}$, we invoke the theorem D.2 on the evolution of $\mathbf{W}\mathbf{W}^\top = \mathbf{U}\mathbf{D}\mathbf{U}^\top$. We ignore it here as it does not have much consequence on our results. $\qquad \square$

**Theorem B.6.** *In the large noise limit, when $l = 2$, the following properties hold, for $t \leq \tau$,*

(a) $\mathbf{s}_0, \mathbf{s}_1$ *are greater than zero almost surely.*

(b) *for $\alpha = (p - 3)/2$, $\mathbf{s}_0^{-\alpha}$ is a super-martingale while $\mathbf{s}_1^{-\alpha}$ is a sub-martingale.*

*Proof.* First, note that in the large noise limit with $l = 2$, the evolution of the eigenvalues is expressed as

$$d(\mathbf{s}_0) = p\mathbf{c}_0^2 dt + \frac{\mathbf{s}_0\mathbf{c}_1^2 + \mathbf{s}_1\mathbf{c}_0^2}{\mathbf{s}_0 - \mathbf{s}_1}dt + 2\sqrt{\mathbf{s}_0\mathbf{c}_0^2}\left(d\tilde{\mathbf{B}}_t\right)_0, \tag{B.7}$$

$$d(\mathbf{s}_1) = p\mathbf{c}_1^2 dt - \frac{\mathbf{s}_0\mathbf{c}_1^2 + \mathbf{s}_1\mathbf{c}_0^2}{\mathbf{s}_0 - \mathbf{s}_1}dt + 2\sqrt{\mathbf{s}_1\mathbf{c}_1^2}\left(d\tilde{\mathbf{B}}_t\right)_1. \tag{B.8}$$

Using the Ito chain rule, for the evolution of $\mathbf{s}_0^{-\alpha}$ we can write

$$d\left(\mathbf{s}_0^{-\alpha}\right) = \frac{\partial\left(\mathbf{s}_0^{-\alpha}\right)}{\partial\mathbf{s}_0}\left(p\mathbf{c}_0^2 dt + \frac{\mathbf{s}_0\mathbf{c}_1^2 + \mathbf{s}_1\mathbf{c}_0^2}{\mathbf{s}_0 - \mathbf{s}_1}dt + 2\sqrt{\mathbf{s}_0\mathbf{c}_0^2}\left(d\tilde{\mathbf{B}}_t\right)_0\right) + \frac{1}{2}\frac{\partial^2\left(\mathbf{s}_0^{-\alpha}\right)}{\partial^2\mathbf{s}_0}\left(2\sqrt{\mathbf{s}_0\mathbf{c}_0^2}\right)^2 dt$$

$$= -\alpha\mathbf{s}_0^{-\alpha-1}\left(p\mathbf{c}_0^2 dt + \frac{\mathbf{s}_0\mathbf{c}_1^2 + \mathbf{s}_1\mathbf{c}_0^2}{\mathbf{s}_0 - \mathbf{s}_1}dt - 2(\alpha+1)\mathbf{c}_0^2 dt + 2\sqrt{\mathbf{s}_0\mathbf{c}_0^2}\left(d\tilde{\mathbf{B}}_t\right)_0\right)$$

$$= -\alpha\mathbf{s}_0^{-\alpha-1}\left(\mathbf{c}_0^2 dt + \frac{\mathbf{s}_0\mathbf{c}_1^2 + \mathbf{s}_1\mathbf{c}_0^2}{\mathbf{s}_0 - \mathbf{s}_1}dt + 2\sqrt{\mathbf{s}_0\mathbf{c}_0^2}\left(d\tilde{\mathbf{B}}_t\right)_0\right),$$

analogously

$$d\left(\mathbf{s}_1^{-\alpha}\right) = -\alpha\mathbf{s}_1^{-\alpha-1}\left(\mathbf{c}_1^2 dt - \frac{\mathbf{s}_0\mathbf{c}_1^2 + \mathbf{s}_1\mathbf{c}_0^2}{\mathbf{s}_0 - \mathbf{s}_1}dt + 2\sqrt{\mathbf{s}_1\mathbf{c}_1^2}\left(d\tilde{\mathbf{B}}_t\right)_1\right),$$

and finally for $\mathbf{s}_0^{-\alpha}\mathbf{s}_1^{-\alpha}$

$$d\left(\mathbf{s}_0^{-\alpha}\mathbf{s}_1^{-\alpha}\right) = d\left(\mathbf{s}_0^{-\alpha}\right)\mathbf{s}_1^{-\alpha} + \mathbf{s}_0^{-\alpha}d\left(\mathbf{s}_1^{-\alpha}\right) + d\left(\mathbf{s}_0^{-\alpha}\right)d\left(\mathbf{s}_1^{-\alpha}\right)$$

$$= -\alpha\mathbf{s}_0^{-\alpha-1}\mathbf{s}_1^{-\alpha}\left(\mathbf{c}_0^2 dt + \frac{\mathbf{s}_0\mathbf{c}_1^2 + \mathbf{s}_1\mathbf{c}_0^2}{\mathbf{s}_0 - \mathbf{s}_1}dt + 2\sqrt{\mathbf{s}_0\mathbf{c}_0^2}\left(d\tilde{\mathbf{B}}_t\right)_0\right)$$

$$-\alpha\mathbf{s}_0^{-\alpha}\mathbf{s}_1^{-\alpha-1}\left(\mathbf{c}_1^2 dt - \frac{\mathbf{s}_0\mathbf{c}_1^2 + \mathbf{s}_1\mathbf{c}_0^2}{\mathbf{s}_0 - \mathbf{s}_1}dt + 2\sqrt{\mathbf{s}_1\mathbf{c}_1^2}\left(d\tilde{\mathbf{B}}_t\right)_1\right).$$

Now, we can show that the drift term in the SDE that describes the dynamics of $\mathbf{s}_0^{-\alpha}\mathbf{s}_1^{-\alpha}$ is zero, which gives us the first part of the result by Mckean's argument [Mayerhofer et al., 2011],

$$-\alpha\mathbf{s}_0^{-\alpha-1}\mathbf{s}_1^{-\alpha-1}\left(\mathbf{s}_1\mathbf{c}_0^2 + \mathbf{s}_1\frac{\mathbf{s}_0\mathbf{c}_1^2 + \mathbf{s}_1\mathbf{c}_0^2}{\mathbf{s}_0 - \mathbf{s}_1} + \mathbf{s}_0\mathbf{c}_1^2 + \mathbf{s}_0\frac{\mathbf{s}_0\mathbf{c}_1^2 + \mathbf{s}_1\mathbf{c}_0^2}{\mathbf{s}_0 - \mathbf{s}_1}\right)$$

$$= -\alpha\mathbf{s}_0^{-\alpha-1}\mathbf{s}_1^{-\alpha-1}\left(\mathbf{s}_1\mathbf{c}_0^2 + \mathbf{s}_0\mathbf{c}_1^2 + \frac{\mathbf{s}_0\mathbf{s}_1\mathbf{c}_1^2 + \mathbf{s}_1^2\mathbf{c}_0^2 - \mathbf{s}_0^2\mathbf{c}_1^2 + \mathbf{s}_0\mathbf{s}_1\mathbf{c}_0^2}{\mathbf{s}_0 - \mathbf{s}_1}\right)$$

$$= -\alpha\mathbf{s}_0^{-\alpha-1}\mathbf{s}_1^{-\alpha-1}\left(\mathbf{s}_1\mathbf{c}_0^2 + \mathbf{s}_0\mathbf{c}_1^2 + \frac{(\mathbf{s}_1 - \mathbf{s}_0)\left(\mathbf{s}_0\mathbf{c}_1^2 + \mathbf{s}_1\mathbf{c}_0^2\right)}{\mathbf{s}_0 - \mathbf{s}_1}\right) = 0.$$

The second part is obtained by noticing that

$$\mathbf{c}_0^2 + \frac{\mathbf{s}_0\mathbf{c}_1^2 + \mathbf{s}_1\mathbf{c}_0^2}{\mathbf{s}_0 - \mathbf{s}_1} = \frac{\mathbf{s}_0\left(\mathbf{c}_1^2 + \mathbf{c}_0^2\right)}{\mathbf{s}_0 - \mathbf{s}_1} \geq 0,$$

$$\mathbf{c}_1^2 - \frac{\mathbf{s}_0\mathbf{c}_1^2 + \mathbf{s}_1\mathbf{c}_0^2}{\mathbf{s}_0 - \mathbf{s}_1} = -\frac{\mathbf{s}_1\left(\mathbf{c}_1^2 + \mathbf{c}_0^2\right)}{\mathbf{s}_0 - \mathbf{s}_1} \leq 0,$$

and hence the drift term of $d\left(\mathbf{s}_0^{-\alpha}\right)$ is not positive, while the drift term of $d\left(\mathbf{s}_1^{-\alpha}\right)$ is not negative.

□

## C Experiment details

In all the graphs we plot the values averaged on 20 runs with different random seeds as well as the 95% confidence interval (lightly colored). To numerically emulate GF (Figure 1), we set a stepsize of $1e^{-6}$ in numerical simulation.

In the further experiments, we study the behaviour of the linear network for regression with the same synthetic data and same network initialization as in previous experiment. As seen in the left plot of the Figure 2, when the stepsize is large ($\eta = 0.1$), singular values exhibit behavior similar to the case of LNGF, while with the small stepsize ($\eta = 0.005$) the evolution of singular values is closer to GF case. Next, we examine the effect of SGD in the case of classification task with logistic loss, as illustrated in the middle plot of the Figure 2. We consider synthetic data with $n = 1000$ samples of Gaussian data in $\mathbb{R}^5$ ($d = 5$) constituting two clusters corresponding to two classes ($k = 1$). Note that larger stepsize ($\eta = 0.5$) in this case also forces the smallest singular value to tend to zero, however the effect is not so dramatic for the rest of singular values. Additionally, we study the 2-layer ReLu network optimized with SGD on the same regression task as before. As seen in the right plot of the Figure 2, the decrease of the last singular value $\sigma_4$ is much slower than in the case of the linear network, however, the larger stepsize still facilitates divergence of $k$ largest ($\sigma_0$ and $\sigma_1$) and $p - k$ smallest ($\sigma_2$, $\sigma_3$ and $\sigma_4$) singular values.

All experiments are implemented with Python 3 [Van Rossum and Drake, 2009] under PSF license, NumPy [Harris et al., 2020] under BSD license, and PyTorch [Paszke et al., 2019] under BSD-3-Clause license.

The experiments were run on a Intel i5-8250U, 8-GB RAM, with OS Ubuntu 20.04.6.

## D Supplementary material

### D.1 Notations and preliminary definitions

**Definition D.1** (Eigen decomposition and Singular Value decomposition). *We discuss the eigen value decomposition for a symmetric square matrix, and the singular value decompostion for any matrix is defined as the following*

(a) ***Eigen decomposition***. *For any rank $r$ matrix $\mathbf{R} \in S_p$, $\mathbf{R} = \mathbf{V}\mathbf{D}\mathbf{V}^\top$ is the eigen decomposition, where $\mathbf{V} \in \mathbb{R}^{p \times r}$, $\mathbf{D} \in \mathbb{R}^{r \times r}$, $\mathbf{D}$ is a diagonal matrix and $\mathbf{V}^\top \mathbf{V} = \mathbf{I}_r$, however, $\mathbf{V}\mathbf{V}^\top$ is not necessarily an identity matrix unless $r = p$.*

(b) ***Singular Value Decomposition***. *For any rank $r$ matrix $\mathbf{W} \in \mathbb{R}^{p \times l}$, $\mathbf{W} = \mathbf{U}\mathbf{\Sigma}\mathbf{V}^\top$, where $\mathbf{U} \in \mathbb{R}^{p \times r}, \mathbf{V} \in \mathbb{R}^{l \times r}, \mathbf{\Sigma} \in \mathbb{R}^{r \times r}$, $\mathbf{\Sigma}$ is a diagonal matrix and $\mathbf{U}^\top \mathbf{U} = \mathbf{V}^\top \mathbf{V} = \mathbf{I}_r$, however the $\mathbf{U}\mathbf{U}^\top$ and $\mathbf{V}\mathbf{V}^\top$ are not necessarily identity unless $r = p$ or $r = l$ respectively.*

### D.2 Eigenvalues of matrix valued stochastic process

**Theorem D.2.** *For a matrix-valued stochastic process on $S_{p+k}^{++}$,*

$$d\mathbf{R} = \mathbf{N}dt + d\mathbf{M}$$

*where $d\mathbf{M}$ is a local martingale process. Let $R = \mathbf{V}\mathbf{D}\mathbf{V}^\top$ is the eigenvalue decomposition of the process, the evolution of eigenvalues satisfy the SDE for time $t$ less than the collision time,*

$$d\mathbf{D} = \mathbf{I} \odot \widetilde{\mathbf{N}}\,dt + \mathbf{I} \odot d\widetilde{\mathbf{M}}\,dt + \mathbf{I} \odot \left(d\widetilde{\mathbf{M}}\left(\mathbf{S} \odot d\widetilde{\mathbf{M}}\right)\right) + \mathbf{D}^{-1} \odot \left(\mathbf{V}^\top d\mathbf{R}\left(\mathbf{I} - \mathbf{V}\mathbf{V}^\top\right)d\mathbf{R}\mathbf{V}\right).$$

*where $\mathbf{S}$ is defined as per Eq. D.1 and $d\widetilde{\mathbf{M}} = \mathbf{V}^\top d\mathbf{M}\mathbf{V}, \widetilde{\mathbf{N}} = \mathbf{V}^\top \mathbf{N}\mathbf{V}$. The evolution of the eigenvectors,*

$$d\mathbf{V} = \mathbf{V}\left(\mathbf{Q}_\parallel\,dt + \mathbf{S} \odot d\mathbf{F}\right) + \left(\mathbf{I} - \mathbf{V}\mathbf{V}^\top\right)\left(\mathbf{Q}_\perp\,dt + d\mathbf{R}\,\mathbf{V}\mathbf{D}^{-1}\right).$$

*where you define,*

$$\mathbf{Q}_{\parallel} = \frac{\mathrm{I} \odot \left[ \left( \mathbf{S} \odot \mathrm{d}\widetilde{\mathbf{M}} \right) \left( \mathbf{S} \odot \mathrm{d}\widetilde{\mathbf{M}} \right) \right]}{2} - \frac{\mathrm{I} \odot \left[ \mathbf{D}^{-1} \mathbf{V}^{\top} \mathrm{d}\mathbf{R} \left( \mathrm{I} - \mathbf{V}\mathbf{V}^{\top} \right) \mathrm{d}\mathbf{R}\mathbf{V}\mathbf{D}^{-1} \right]}{2}$$
$$- \mathbf{S} \odot \left[ \left( \mathbf{S} \odot \mathrm{d}\widetilde{\mathbf{M}} \right) \left[ \mathrm{d}\widetilde{\mathbf{M}} \odot \mathrm{I} \right] \right] + \mathbf{S} \odot \left( \mathrm{d}\widetilde{\mathbf{M}} \left( \mathbf{S} \odot \mathrm{d}\widetilde{\mathbf{M}} \right) \right)$$
$$+ \mathbf{S} \odot \left( \mathbf{V}^{\top} \mathrm{d}\mathbf{R} \left( \mathrm{I} - \mathbf{V}\mathbf{V}^{\top} \right) \mathrm{d}\mathbf{R}\mathbf{V}\mathbf{D}^{-1} \right),$$
$$\mathbf{Q}_{\perp} = \left[ \mathrm{d}\mathbf{R}\mathbf{V}\mathbf{D}^{-1} \right] \left[ \left[ \mathbf{S} \odot \mathrm{d}\widetilde{\mathbf{M}} \right] \mathbf{D} - \mathrm{d}\widetilde{\mathbf{M}} \right] \mathbf{D}^{-1}.$$

**Evolution of eigenvalues for general matrix SDE**

*Proof.* Using the eigen decomposition, we have $\mathbf{R} = \mathbf{V}\mathbf{D}\mathbf{V}^{\top}$,

$$\mathbf{D} = \mathbf{V}^{\top}\mathbf{R}\mathbf{V},$$
$$\mathrm{d}\mathbf{D} = \mathbf{V}^{\top}\mathrm{d}\mathbf{R}\mathbf{V} + \mathbf{V}^{\top}\mathbf{R}\mathrm{d}\mathbf{V} + \mathrm{d}\mathbf{V}^{\top}\mathbf{R}\mathbf{V} + \mathbf{V}^{\top}\mathrm{d}\mathbf{R}\mathrm{d}\mathbf{V} + \mathrm{d}\mathbf{V}^{\top}\mathrm{d}\mathbf{R}\mathbf{V} + \mathrm{d}\mathbf{V}^{\top}\mathbf{R}\mathrm{d}\mathbf{V},$$
$$= \mathbf{V}^{\top}\mathrm{d}\mathbf{R}\mathbf{V} + \mathbf{D}\mathbf{V}^{\top}\mathrm{d}\mathbf{V} + \mathrm{d}\mathbf{V}^{\top}\mathbf{V}\mathbf{D} + \mathbf{V}^{\top}\mathrm{d}\mathbf{R}\mathrm{d}\mathbf{V} + \mathrm{d}\mathbf{V}^{\top}\mathrm{d}\mathbf{R}\mathbf{V} + \left( \mathrm{d}\mathbf{V}^{\top}\mathbf{V} \right) \mathbf{D} \left( \mathbf{V}^{\top}\mathrm{d}\mathbf{V} \right).$$

The approach we follow is use the jacobian of the evolution of $\mathbf{V}$ (see [Townsend, 2016] ) and solve the constrains equations to obtain the Ito correction term as done in Bru [1989]. Let $(\mathbf{s}_1, \mathbf{s}_2, \ldots, \mathbf{s}_r)$ denote the diagonal entries of $\mathbf{D}$. Furthermore, we define the matrix $\mathbf{S}$, which plays a notable role in Jacobian w.r.t $\mathbf{V}$, as the following,

$$\mathbf{S}_{ij} = \begin{cases} 0 & \text{if } i = j, \\ (\mathbf{s}_j - \mathbf{s}_i)^{-1} & \text{o.w.} \end{cases} \tag{D.1}$$

For the sake of brevity, we denote the evolution

$$\mathrm{d}\mathbf{F} \stackrel{\text{def}}{=} \mathbf{V}^{\top}\mathrm{d}\mathbf{R}\mathbf{V} = \mathbf{V}^{\top}\mathbf{N}\mathbf{V}\,\mathrm{d}t + \mathbf{V}^{\top}\mathrm{d}\mathbf{M}\mathbf{V},$$
$$\stackrel{\text{def}}{=} \widetilde{\mathbf{N}}\,\mathrm{d}t + \mathrm{d}\widetilde{\mathbf{M}}$$

The evolution of the eigenvectors,

$$\mathrm{d}\mathbf{V} = \mathbf{V}\mathrm{d}\Omega_{\mathbf{V}} + (\mathrm{I} - \mathbf{V}\mathbf{V}^{\top})\mathrm{d}\Xi_{\mathbf{V}}.$$

Using the Jacobian of the eigen vectors, we write,

$$\mathrm{d}\Omega_{\mathbf{V}} = \mathbf{Q}_{\parallel}\,\mathrm{d}t + \mathbf{S} \odot \mathrm{d}\mathbf{F},$$
$$\mathrm{d}\Xi_{\mathbf{V}} = \mathbf{Q}_{\perp}\,\mathrm{d}t + \mathrm{d}\mathbf{R}\,\mathbf{V}\mathbf{D}^{-1}.$$

Note that $\mathbf{V}^{\top}\mathbf{V} = \mathrm{I}_r$, using this we have,

$$0 = \mathrm{d}\left( \mathbf{V}^{\top}\mathbf{V} \right) = \mathrm{d}\mathbf{V}^{\top}\mathbf{V} + \mathbf{V}^{\top}\mathrm{d}\mathbf{V} + \mathrm{d}\mathbf{V}^{\top}\mathrm{d}\mathbf{V},$$
$$= \mathrm{d}\Omega_{\mathbf{V}}^{\top} + \mathrm{d}\Omega_{\mathbf{V}} + \mathrm{d}\mathbf{V}^{\top}\mathbf{V}\mathbf{V}^{\top}\mathrm{d}\mathbf{V} + \mathrm{d}\mathbf{V}^{\top} \left( \mathrm{I} - \mathbf{V}\mathbf{V}^{\top} \right) \mathrm{d}\mathbf{V},$$
$$= \mathrm{d}\Omega_{\mathbf{V}}^{\top} + \mathrm{d}\Omega_{\mathbf{V}} + \mathrm{d}\Omega_{\mathbf{V}}^{\top}\mathrm{d}\Omega_{\mathbf{V}} + \mathrm{d}\Xi_{\mathbf{V}}^{\top} \left( \mathrm{I} - \mathbf{V}\mathbf{V}^{\top} \right) \mathrm{d}\Xi_{\mathbf{V}},$$
$$= \mathrm{d}\Omega_{\mathbf{V}}^{\top} + \mathrm{d}\Omega_{\mathbf{V}} - (\mathbf{S} \odot \mathrm{d}\mathbf{F})(\mathbf{S} \odot \mathrm{d}\mathbf{F}) + \mathbf{D}^{-1}\mathbf{V}^{\top}\mathrm{d}\mathbf{R} \left( \mathrm{I} - \mathbf{V}\mathbf{V}^{\top} \right) \mathrm{d}\mathbf{R}\mathbf{V}\mathbf{D}^{-1}.$$

Using $\mathrm{d}\Omega_{\mathbf{V}}^{\top} = \mathbf{Q}_{\parallel}^{\top}\mathrm{d}t - \mathbf{S} \odot \mathrm{d}\mathbf{F}$, we have $\mathrm{d}\Omega_{\mathbf{V}}^{\top} + \mathrm{d}\Omega_{\mathbf{V}} = \left( \mathbf{Q}_{\parallel}^{\top} + \mathbf{Q}_{\parallel} \right) \mathrm{d}t$.

$$\left( \mathbf{Q}_{\parallel} + \mathbf{Q}_{\parallel}^{\top} \right) \mathrm{d}t = \left( \mathbf{S} \odot \mathrm{d}\widetilde{\mathbf{M}} \right) \left( \mathbf{S} \odot \mathrm{d}\widetilde{\mathbf{M}} \right) - \mathbf{D}^{-1}\mathbf{V}^{\top}\mathrm{d}\mathbf{R} \left( \mathrm{I} - \mathbf{V}\mathbf{V}^{\top} \right) \mathrm{d}\mathbf{R}\mathbf{V}\mathbf{D}^{-1}. \tag{D.2}$$

Coming back to the evolution of singular values,

$$\mathrm{d}\mathbf{D} = \mathbf{V}^{\top}\mathrm{d}\mathbf{R}\mathbf{V} + \mathbf{D}\mathbf{V}^{\top}\mathrm{d}\mathbf{V} + \mathrm{d}\mathbf{V}^{\top}\mathbf{V}\mathbf{D} + \mathbf{V}^{\top}\mathrm{d}\mathbf{R}\mathrm{d}\mathbf{V} + \mathrm{d}\mathbf{V}^{\top}\mathrm{d}\mathbf{R}\mathbf{V} + \left( \mathrm{d}\mathbf{V}^{\top}\mathbf{V} \right) \mathbf{D} \left( \mathbf{V}^{\top}\mathrm{d}\mathbf{V} \right).$$
$$= \mathrm{d}\mathbf{F} + \left( \mathbf{D}\mathbf{Q}_{\parallel} + \mathbf{Q}_{\parallel}^{\top}\mathbf{D} \right) \mathrm{d}t + \mathbf{D} \left( \mathbf{S} \odot \mathrm{d}\mathbf{F} \right) - \left( \mathbf{S} \odot \mathrm{d}\mathbf{F} \right) \mathbf{D} + \mathrm{d}\Omega_{\mathbf{V}}^{\top}\mathbf{D}\mathrm{d}\Omega_{\mathbf{V}}$$
$$+ \mathbf{V}^{\top}\mathrm{d}\mathbf{R} \left[ \mathbf{V}\mathrm{d}\Omega_{\mathbf{V}} + \left( \mathrm{I} - \mathbf{V}\mathbf{V}^{\top} \right) \mathrm{d}\Xi_{\mathbf{V}} \right] + \left[ \mathrm{d}\Omega_{\mathbf{V}}^{\top}\mathbf{V}^{\top} + \mathrm{d}\Xi_{\mathbf{V}}^{\top} \left( \mathrm{I} - \mathbf{V}\mathbf{V}^{\top} \right) \right] \mathrm{d}\mathbf{R}\mathbf{V},$$

$$\mathrm{d}\mathbf{D} = \mathrm{I} \odot \mathrm{d}\mathbf{F} + \left(\mathbf{D}\mathbf{Q}_{\parallel} + \mathbf{Q}_{\parallel}^{\top}\mathbf{D}\right)\mathrm{d}t - \left(\mathbf{S} \odot \mathrm{d}\widetilde{\mathbf{M}}\right)\mathbf{D}\left(\mathbf{S} \odot \mathrm{d}\widetilde{\mathbf{M}}\right) + \mathrm{d}\widetilde{\mathbf{M}}\left(\mathbf{S} \odot \mathrm{d}\widetilde{\mathbf{M}}\right)$$
$$- \left(\mathbf{S} \odot \mathrm{d}\widetilde{\mathbf{M}}\right)\mathrm{d}\widetilde{\mathbf{M}} + \mathbf{V}^{\top}\mathrm{d}\mathbf{R}\left(\mathrm{I} - \mathbf{V}\mathbf{V}^{\top}\right)\mathrm{d}\mathbf{R}\mathbf{V}\mathbf{D}^{-1} + \mathbf{D}^{-1}\mathbf{V}^{\top}\mathrm{d}\mathbf{R}\left(\mathrm{I} - \mathbf{V}\mathbf{V}^{\top}\right)\mathrm{d}\mathbf{R}. \tag{D.3}$$

Note that $\mathrm{d}\mathbf{D}$ is diagonal, hence, $\mathrm{I} \odot \mathrm{d}\mathbf{D} = \mathrm{d}\mathbf{D}$.

$$\mathrm{I} \odot \mathrm{d}\mathbf{D} = \mathrm{I} \odot \mathrm{d}\mathbf{F} + \mathrm{I} \odot \left(\mathbf{D}\mathbf{Q}_{\parallel} + \mathbf{Q}_{\parallel}^{\top}\mathbf{D}\right)\mathrm{d}t - \mathrm{I} \odot \left[\left(\mathbf{S} \odot \mathrm{d}\widetilde{\mathbf{M}}\right)\mathbf{D}\left(\mathbf{S} \odot \mathrm{d}\widetilde{\mathbf{M}}\right)\right]$$
$$+ 2\mathrm{I} \odot \left(\mathrm{d}\widetilde{\mathbf{M}}\left(\mathbf{S} \odot \mathrm{d}\widetilde{\mathbf{M}}\right)\right) + 2\mathrm{I} \odot \left(\mathbf{D}^{-1}\mathbf{V}^{\top}\mathrm{d}\mathbf{R}\left(\mathrm{I} - \mathbf{V}\mathbf{V}^{\top}\right)\mathrm{d}\mathbf{R}\right)$$

Note that $\mathrm{I} \odot (DM) = \mathrm{I} \odot (MD) = D \odot M$ for any matrix $M$ and diagonal matrix $D$, using this property, we can simplify the above expression as,

$$\mathrm{d}\mathbf{D} = \mathrm{I} \odot \mathrm{d}\mathbf{F} + \mathbf{D} \odot \left(\mathbf{Q}_{\parallel} + \mathbf{Q}_{\parallel}^{\top}\right)\mathrm{d}t - \mathrm{I} \odot \left[\left(\mathbf{S} \odot \mathrm{d}\widetilde{\mathbf{M}}\right)\mathbf{D}\left(\mathbf{S} \odot \mathrm{d}\widetilde{\mathbf{M}}\right)\right]$$
$$+ 2\mathrm{I} \odot \left(\mathrm{d}\widetilde{\mathbf{M}}\left(\mathbf{S} \odot \mathrm{d}\widetilde{\mathbf{M}}\right)\right) + 2\mathbf{D}^{-1} \odot \left(\mathbf{V}^{\top}\mathrm{d}\mathbf{R}\left(\mathrm{I} - \mathbf{V}\mathbf{V}^{\top}\right)\mathrm{d}\mathbf{R}\right)$$

Using Eq. D.2, we have,

$$\mathbf{D} \odot \left(\mathbf{Q}_{\parallel} + \mathbf{Q}_{\parallel}^{\top}\right)\mathrm{d}t = \mathbf{D} \odot \left[\left(\mathbf{S} \odot \mathrm{d}\widetilde{\mathbf{M}}\right)\left(\mathbf{S} \odot \mathrm{d}\widetilde{\mathbf{M}}\right) - \mathbf{D}^{-1}\mathbf{V}^{\top}\mathrm{d}\mathbf{R}\left(\mathrm{I} - \mathbf{V}\mathbf{V}^{\top}\right)\mathrm{d}\mathbf{R}\mathbf{V}\mathbf{D}^{-1}\right],$$
$$= \mathrm{I} \odot \left[\left(\mathbf{S} \odot \mathrm{d}\widetilde{\mathbf{M}}\right)\left(\mathbf{S} \odot \mathrm{d}\widetilde{\mathbf{M}}\right)\mathbf{D}\right] - \mathbf{D}^{-1} \odot \left(\mathbf{V}^{\top}\mathrm{d}\mathbf{R}\left(\mathrm{I} - \mathbf{V}\mathbf{V}^{\top}\right)\mathrm{d}\mathbf{R}\mathbf{V}\right).$$

Using this,

$$\mathrm{d}\mathbf{D} = \mathrm{I} \odot \mathrm{d}\mathbf{F} + \mathrm{I} \odot \left[\left(\mathbf{S} \odot \mathrm{d}\widetilde{\mathbf{M}}\right)\left(\mathbf{S} \odot \mathrm{d}\widetilde{\mathbf{M}}\right)\mathbf{D}\right] - \mathrm{I} \odot \left[\left(\mathbf{S} \odot \mathrm{d}\widetilde{\mathbf{M}}\right)\mathbf{D}\left(\mathbf{S} \odot \mathrm{d}\widetilde{\mathbf{M}}\right)\right]$$
$$+ 2\mathrm{I} \odot \left(\mathrm{d}\widetilde{\mathbf{M}}\left(\mathbf{S} \odot \mathrm{d}\widetilde{\mathbf{M}}\right)\right) + \mathbf{D}^{-1} \odot \left(\mathbf{V}^{\top}\mathrm{d}\mathbf{R}\left(\mathrm{I} - \mathbf{V}\mathbf{V}^{\top}\right)\mathrm{d}\mathbf{R}\mathbf{V}\right),$$
$$= \mathrm{I} \odot \mathrm{d}\mathbf{F} + \mathrm{I} \odot \left[\left(\mathbf{S} \odot \mathrm{d}\widetilde{\mathbf{M}}\right)\left[\left(\mathbf{S} \odot \mathrm{d}\widetilde{\mathbf{M}}\right)\mathbf{D} - \mathbf{D}\left(\mathbf{S} \odot \mathrm{d}\widetilde{\mathbf{M}}\right)\right]\right]$$
$$+ 2\mathrm{I} \odot \left(\mathrm{d}\widetilde{\mathbf{M}}\left(\mathbf{S} \odot \mathrm{d}\widetilde{\mathbf{M}}\right)\right) + \mathbf{D}^{-1} \odot \left(\mathbf{V}^{\top}\mathrm{d}\mathbf{R}\left(\mathrm{I} - \mathbf{V}\mathbf{V}^{\top}\right)\mathrm{d}\mathbf{R}\mathbf{V}\right),$$
$$= \mathrm{I} \odot \mathrm{d}\mathbf{F} + \mathrm{I} \odot \left[\left(\mathbf{S} \odot \mathrm{d}\widetilde{\mathbf{M}}\right)\mathrm{d}\widetilde{\mathbf{M}}\right]$$
$$+ 2\mathrm{I} \odot \left(\mathrm{d}\widetilde{\mathbf{M}}\left(\mathbf{S} \odot \mathrm{d}\widetilde{\mathbf{M}}\right)\right) + \mathbf{D}^{-1} \odot \left(\mathbf{V}^{\top}\mathrm{d}\mathbf{R}\left(\mathrm{I} - \mathbf{V}\mathbf{V}^{\top}\right)\mathrm{d}\mathbf{R}\mathbf{V}\right),$$
$$= \mathrm{I} \odot \mathrm{d}\mathbf{F} + \mathrm{I} \odot \left(\mathrm{d}\widetilde{\mathbf{M}}\left(\mathbf{S} \odot \mathrm{d}\widetilde{\mathbf{M}}\right)\right) + \mathbf{D}^{-1} \odot \left(\mathbf{V}^{\top}\mathrm{d}\mathbf{R}\left(\mathrm{I} - \mathbf{V}\mathbf{V}^{\top}\right)\mathrm{d}\mathbf{R}\mathbf{V}\right).$$

**Evolution of eigenvectors for general matrix SDE.** Here, we derive the evolution of eigenvectors,

Using Eq. D.2, we have,

$$\left(\mathbf{Q}_{\parallel}\mathbf{D} + \mathbf{Q}_{\parallel}^{\top}\mathbf{D}\right)\mathrm{d}t = \left(\mathbf{S} \odot \mathrm{d}\widetilde{\mathbf{M}}\right)\left(\mathbf{S} \odot \mathrm{d}\widetilde{\mathbf{M}}\right)\mathbf{D} - \mathbf{D}^{-1}\mathbf{V}^{\top}\mathrm{d}\mathbf{R}\left(\mathrm{I} - \mathbf{V}\mathbf{V}^{\top}\right)\mathrm{d}\mathbf{R}\mathbf{V}$$

Now further using the constrain that $\mathrm{d}\mathbf{D}$ needs to be diagonal we get,

$$\left(\mathbf{D}\mathbf{Q}_{\parallel} + \mathbf{Q}_{\parallel}^{\top}\mathbf{D}\right)\mathrm{d}t = \mathrm{d}\mathbf{D} - \mathrm{I} \odot \mathrm{d}\mathbf{F} + \left(\mathbf{S} \odot \mathrm{d}\widetilde{\mathbf{M}}\right)\mathbf{D}\left(\mathbf{S} \odot \mathrm{d}\widetilde{\mathbf{M}}\right) - \mathrm{d}\widetilde{\mathbf{M}}\left(\mathbf{S} \odot \mathrm{d}\widetilde{\mathbf{M}}\right) + \left(\mathbf{S} \odot \mathrm{d}\widetilde{\mathbf{M}}\right)\mathrm{d}\widetilde{\mathbf{M}}$$
$$- \mathbf{V}^{\top}\mathrm{d}\mathbf{R}\left(\mathrm{I} - \mathbf{V}\mathbf{V}^{\top}\right)\mathrm{d}\mathbf{R}\mathbf{V}\mathbf{D}^{-1} - \mathbf{D}^{-1}\mathbf{V}^{\top}\mathrm{d}\mathbf{R}\left(\mathrm{I} - \mathbf{V}\mathbf{V}^{\top}\right)\mathrm{d}\mathbf{R}.$$
$$\left(\mathbf{D}\mathbf{Q}_{\parallel} - \mathbf{Q}_{\parallel}\mathbf{D}\right)\mathrm{d}t = \mathrm{d}\mathbf{D} - \mathrm{I} \odot \mathrm{d}\mathbf{F} - \left(\mathbf{S} \odot \mathrm{d}\widetilde{\mathbf{M}}\right)\left[\left(\mathbf{S} \odot \mathrm{d}\widetilde{\mathbf{M}}\right)\mathbf{D} - \mathbf{D}\left(\mathbf{S} \odot \mathrm{d}\widetilde{\mathbf{M}}\right)\right] - \mathrm{d}\widetilde{\mathbf{M}}\left(\mathbf{S} \odot \mathrm{d}\widetilde{\mathbf{M}}\right)$$
$$+ \left(\mathbf{S} \odot \mathrm{d}\widetilde{\mathbf{M}}\right)\mathrm{d}\widetilde{\mathbf{M}} - \mathbf{V}^{\top}\mathrm{d}\mathbf{R}\left(\mathrm{I} - \mathbf{V}\mathbf{V}^{\top}\right)\mathrm{d}\mathbf{R}\mathbf{V}\mathbf{D}^{-1},$$
$$= \mathrm{d}\mathbf{D} - \mathrm{I} \odot \mathrm{d}\mathbf{F} - \left(\mathbf{S} \odot \mathrm{d}\widetilde{\mathbf{M}}\right)\left[\mathrm{d}\widetilde{\mathbf{M}} \odot \bar{\mathrm{I}}\right] - \mathrm{d}\widetilde{\mathbf{M}}\left(\mathbf{S} \odot \mathrm{d}\widetilde{\mathbf{M}}\right)$$
$$+ \left(\mathbf{S} \odot \mathrm{d}\widetilde{\mathbf{M}}\right)\mathrm{d}\widetilde{\mathbf{M}} - \mathbf{V}^{\top}\mathrm{d}\mathbf{R}\left(\mathrm{I} - \mathbf{V}\mathbf{V}^{\top}\right)\mathrm{d}\mathbf{R}\mathbf{V}\mathbf{D}^{-1},$$
$$= \mathrm{d}\mathbf{D} - \mathrm{I} \odot \mathrm{d}\mathbf{F} + \left(\mathbf{S} \odot \mathrm{d}\widetilde{\mathbf{M}}\right)\left[\mathrm{d}\widetilde{\mathbf{M}} \odot \mathrm{I}\right] - \mathrm{d}\widetilde{\mathbf{M}}\left(\mathbf{S} \odot \mathrm{d}\widetilde{\mathbf{M}}\right)$$
$$- \mathbf{V}^{\top}\mathrm{d}\mathbf{R}\left(\mathrm{I} - \mathbf{V}\mathbf{V}^{\top}\right)\mathrm{d}\mathbf{R}\mathbf{V}\mathbf{D}^{-1}.$$
$$\bar{\mathrm{I}} \odot \left(\mathbf{D}\mathbf{Q}_{\parallel} - \mathbf{Q}_{\parallel}\mathbf{D}\right)\mathrm{d}t = \bar{\mathrm{I}} \odot \left(\mathrm{d}\mathbf{D} - \mathrm{I} \odot \mathrm{d}\mathbf{F}\right) + \bar{\mathrm{I}} \odot \left[\left(\mathbf{S} \odot \mathrm{d}\widetilde{\mathbf{M}}\right)\left[\mathrm{d}\widetilde{\mathbf{M}} \odot \mathrm{I}\right]\right] - \bar{\mathrm{I}} \odot \left(\mathrm{d}\widetilde{\mathbf{M}}\left(\mathbf{S} \odot \mathrm{d}\widetilde{\mathbf{M}}\right)\right)$$
$$- \bar{\mathrm{I}} \odot \left(\mathbf{V}^{\top}\mathrm{d}\mathbf{R}\left(\mathrm{I} - \mathbf{V}\mathbf{V}^{\top}\right)\mathrm{d}\mathbf{R}\mathbf{V}\mathbf{D}^{-1}\right).$$

$$\left(\bar{I} \odot \mathbf{Q}_\parallel\right) dt = \mathbf{S} \odot \left[ -\left(\mathbf{S} \odot d\widetilde{\mathbf{M}}\right)\left[d\widetilde{\mathbf{M}} \odot I\right] + d\widetilde{\mathbf{M}}\left(\mathbf{S} \odot d\widetilde{\mathbf{M}}\right) + \mathbf{V}^\top d\mathbf{R}\left(I - \mathbf{V}\mathbf{V}^\top\right)d\mathbf{R}\mathbf{V}\mathbf{D}^{-1}\right].$$

Combing these, we get the diagonal and off diagonal terms of $\mathbf{Q}_\parallel$

$$\left(I \odot \mathbf{Q}_\parallel\right) dt = \frac{1}{2} I \odot \left(\mathbf{Q}_\parallel + \mathbf{Q}_\parallel^\top\right) dt,$$

$$= \frac{I \odot \left[\left(\mathbf{S} \odot d\widetilde{\mathbf{M}}\right)\left(\mathbf{S} \odot d\widetilde{\mathbf{M}}\right)\right]}{2} - \frac{I \odot \left[\mathbf{D}^{-1}\mathbf{V}^\top d\mathbf{R}\left(I - \mathbf{V}\mathbf{V}^\top\right)d\mathbf{R}\mathbf{V}\mathbf{D}^{-1}\right]}{2}.$$

$$\mathbf{Q}_\parallel = \frac{I \odot \left[\left(\mathbf{S} \odot d\widetilde{\mathbf{M}}\right)\left(\mathbf{S} \odot d\widetilde{\mathbf{M}}\right)\right]}{2} - \frac{I \odot \left[\mathbf{D}^{-1}\mathbf{V}^\top d\mathbf{R}\left(I - \mathbf{V}\mathbf{V}^\top\right)d\mathbf{R}\mathbf{V}\mathbf{D}^{-1}\right]}{2}$$
$$- \mathbf{S} \odot \left[\left(\mathbf{S} \odot d\widetilde{\mathbf{M}}\right)\left[d\widetilde{\mathbf{M}} \odot I\right]\right] + \mathbf{S} \odot \left(d\widetilde{\mathbf{M}}\left(\mathbf{S} \odot d\widetilde{\mathbf{M}}\right)\right)$$
$$+ \mathbf{S} \odot \left(\mathbf{V}^\top d\mathbf{R}\left(I - \mathbf{V}\mathbf{V}^\top\right)d\mathbf{R}\mathbf{V}\mathbf{D}^{-1}\right).$$

**Computing of $\mathbf{Q}_\perp$.** Recalling the evolution of the eigenvectors,
$$d\mathbf{V} = \mathbf{V}d\Omega_\mathbf{V} + (I - \mathbf{V}\mathbf{V}^\top)d\Xi_\mathbf{V}.$$

Using the Jacobian of the eigen vectors, we write,
$$d\Omega_\mathbf{V} = \mathbf{Q}_\parallel\, dt + \mathbf{S} \odot d\mathbf{F},$$
$$d\Xi_\mathbf{V} = \mathbf{Q}_\perp\, dt + d\mathbf{R}\, \mathbf{V}\mathbf{D}^{-1},$$
$$d\mathbf{V} = \mathbf{V}\left[\mathbf{Q}_\parallel\, dt + \mathbf{S} \odot d\mathbf{F}\right] + (I - \mathbf{V}\mathbf{V}^\top)\left[\mathbf{Q}_\perp\, dt + d\mathbf{R}\, \mathbf{V}\mathbf{D}^{-1}\right],$$
$$d\mathbf{V}^\top = \left[\mathbf{Q}_\parallel^\top\, dt - \mathbf{S} \odot d\mathbf{F}\right]\mathbf{V}^\top + \left[\mathbf{Q}_\perp^\top dt + \mathbf{D}^{-1}\mathbf{V}^\top d\mathbf{R}\right](I - \mathbf{V}\mathbf{V}^\top).$$

Using the fact that $\left(I - \mathbf{V}\mathbf{V}^\top\right)\mathbf{R} = 0$ and deriving it,
$$0 = \left(I - \mathbf{V}\mathbf{V}^\top\right)\mathbf{R},$$
$$0 = d\left[\left(I - \mathbf{V}\mathbf{V}^\top\right)\mathbf{R}\right],$$
$$d\mathbf{R} = d\left(\mathbf{V}\mathbf{V}^\top\mathbf{R}\right),$$
$$= d\mathbf{V}\mathbf{V}^\top\mathbf{R} + \mathbf{V}d\mathbf{V}^\top\mathbf{R} + \mathbf{V}\mathbf{V}^\top d\mathbf{R} + d\mathbf{V}d\mathbf{V}^\top\mathbf{R} + d\mathbf{V}\mathbf{V}^\top d\mathbf{R} + \mathbf{V}d\mathbf{V}^\top d\mathbf{R},$$
$$d\mathbf{R}\mathbf{V} = d\mathbf{V}\mathbf{D} + \mathbf{V}d\mathbf{V}^\top\mathbf{V}\mathbf{D} + \mathbf{V}\mathbf{V}^\top d\mathbf{R}\mathbf{V} + d\mathbf{V}d\mathbf{V}^\top\mathbf{V}\mathbf{D} + d\mathbf{V}\mathbf{V}^\top d\mathbf{R}\mathbf{V} + \mathbf{V}d\mathbf{V}^\top d\mathbf{R}\mathbf{V},$$

$$d\mathbf{V}\mathbf{D} = \mathbf{V}\left[\mathbf{Q}_\parallel\mathbf{D}\, dt + (\mathbf{S} \odot d\mathbf{F})\,\mathbf{D}\right] + (I - \mathbf{V}\mathbf{V}^\top)\left[\mathbf{Q}_\perp\mathbf{D}\, dt + d\mathbf{R}\,\mathbf{V}\right],$$
$$\mathbf{V}d\mathbf{V}^\top\mathbf{V}\mathbf{D} = \mathbf{V}\left[\mathbf{Q}_\parallel^\top\, dt - \mathbf{S} \odot d\mathbf{F}\right]\mathbf{D},$$
$$d\mathbf{V}d\mathbf{V}^\top\mathbf{V}\mathbf{D} = -\mathbf{V}\left[\mathbf{S} \odot d\mathbf{F}\right]\left[\mathbf{S} \odot d\mathbf{F}\right]\mathbf{D} - \left[(I - \mathbf{V}\mathbf{V}^\top)d\mathbf{R}\mathbf{V}\mathbf{D}^{-1}\right]\left[\mathbf{S} \odot d\mathbf{F}\right]\mathbf{D},$$
$$d\mathbf{V}\mathbf{V}^\top d\mathbf{R}\mathbf{V} = \mathbf{V}\left[\mathbf{S} \odot d\mathbf{F}\right]d\mathbf{F} + \left[(I - \mathbf{V}\mathbf{V}^\top)d\mathbf{R}\mathbf{V}\mathbf{D}^{-1}\right]d\mathbf{F},$$
$$\mathbf{V}d\mathbf{V}^\top d\mathbf{R}\mathbf{V} = -\mathbf{V}\left[\mathbf{S} \odot d\mathbf{F}\right]d\mathbf{F} + \mathbf{V}\mathbf{D}^{-1}\mathbf{V}^\top d\mathbf{R}(I - \mathbf{V}\mathbf{V}^\top)d\mathbf{R}\mathbf{V}.$$

Adding the terms up we get,
$$\mathbf{V}\left[\mathbf{Q}_\parallel + \mathbf{Q}_\parallel^\top\right]\mathbf{D}dt + \left(I - \mathbf{V}\mathbf{V}^\top\right)\mathbf{Q}_\perp\mathbf{D}dt$$
$$- \mathbf{V}\left[\mathbf{S} \odot d\mathbf{F}\right]\left[\mathbf{S} \odot d\mathbf{F}\right]\mathbf{D} - \left[(I - \mathbf{V}\mathbf{V}^\top)d\mathbf{R}\mathbf{V}\mathbf{D}^{-1}\right]\left[\mathbf{S} \odot d\mathbf{F}\right]\mathbf{D}$$
$$+ \left[(I - \mathbf{V}\mathbf{V}^\top)d\mathbf{R}\mathbf{V}\mathbf{D}^{-1}\right]d\mathbf{F} + \mathbf{V}\mathbf{D}^{-1}\mathbf{V}^\top d\mathbf{R}(I - \mathbf{V}\mathbf{V}^\top)d\mathbf{R}\mathbf{V} = 0.$$

$$\left(I - \mathbf{V}\mathbf{V}^\top\right)\mathbf{Q}_\perp\mathbf{D}dt - \left[(I - \mathbf{V}\mathbf{V}^\top)d\mathbf{R}\mathbf{V}\mathbf{D}^{-1}\right]\left[\mathbf{S} \odot d\mathbf{F}\right]\mathbf{D} + \left[(I - \mathbf{V}\mathbf{V}^\top)d\mathbf{R}\mathbf{V}\mathbf{D}^{-1}\right]d\mathbf{F} = 0.$$

$$\left(I - \mathbf{V}\mathbf{V}^\top\right)\mathbf{Q}_\perp\mathbf{D}dt = \left[(I - \mathbf{V}\mathbf{V}^\top)d\mathbf{R}\mathbf{V}\mathbf{D}^{-1}\right]\left[\mathbf{S} \odot d\mathbf{F}\right]\mathbf{D} - \left[(I - \mathbf{V}\mathbf{V}^\top)d\mathbf{R}\mathbf{V}\mathbf{D}^{-1}\right]d\mathbf{F},$$
$$\left(I - \mathbf{V}\mathbf{V}^\top\right)\mathbf{Q}_\perp = \left[(I - \mathbf{V}\mathbf{V}^\top)d\mathbf{R}\mathbf{V}\mathbf{D}^{-1}\right]\left[[\mathbf{S} \odot d\mathbf{F}]\mathbf{D} - d\mathbf{F}\right]\mathbf{D}^{-1}$$

$$\mathbf{Q}_\perp = \left[d\mathbf{R}\mathbf{V}\mathbf{D}^{-1}\right]\left[[\mathbf{S} \odot d\mathbf{F}]\mathbf{D} - d\mathbf{F}\right]\mathbf{D}^{-1}$$

This gives the expression for $\mathbf{Q}_\perp$ and this ends our computation. $\qquad\square$

**Lemma D.3.** *For any matrix $A \in \mathbb{R}^{n \times m}, B \in \mathbb{R}^{n \times n}$, $m \times n$-dimensional Brownian motion $\mathrm{d}\mathbf{B}_t$, the following results hold on the covariance*

$$\mathrm{d}\mathbf{B}_t A \mathrm{d}\mathbf{B}_t = A^\top \mathrm{d}t, \tag{D.4}$$

$$\mathrm{d}\mathbf{B}_t B \mathrm{d}\mathbf{B}_t^\top = \mathrm{tr}\,(B)\,\mathrm{I}_m \mathrm{d}t. \tag{D.5}$$

**Lemma D.4.** *With $\mathbf{S}$ defined in Equation* (D.1)*, $\mathrm{d}\mathbf{F} = \mathrm{d}\mathbf{F} = \boldsymbol{\Sigma}\mathbf{V}^\top \mathrm{d}\mathbf{B}_t \mathbf{c}^\top + \mathbf{c}\mathrm{d}\mathbf{B}_t^\top \mathbf{V}\boldsymbol{\Sigma}$ and $\mathrm{d}\mathbf{m}_t \stackrel{\mathrm{def}}{=} (\boldsymbol{\sigma} \odot \mathrm{d}\tilde{\mathbf{B}}_t)$.*

$$\mathrm{d}\mathbf{F}(\mathbf{S} \odot \mathrm{d}\mathbf{F}) = \mathbf{c}\mathbf{s}^\top \mathbf{S}\mathrm{diag}\,(\mathbf{c})\,\mathrm{d}t - \mathbf{D}\mathrm{diag}\,(\mathbf{S}\mathrm{diag}\,(\mathbf{c})\,\mathbf{c})\,\mathrm{d}t + \mathbf{D}\mathrm{diag}\,(\mathbf{c})\,\mathbf{S}\mathrm{diag}\,(\mathbf{c})\,\mathrm{d}t. \tag{D.6}$$

*Proof.*

$$\mathbf{S} \odot \mathrm{d}\mathbf{F} = [\mathrm{diag}\,(\mathbf{c})\,\mathbf{S}\mathrm{diag}\,(\mathrm{d}\mathbf{m}_t) + \mathrm{diag}\,(\mathrm{d}\mathbf{m}_t)\,\mathbf{S}\mathrm{diag}\,(\mathbf{c})],$$

$$\mathrm{d}\mathbf{F}(\mathbf{S} \odot \mathrm{d}\mathbf{F}) = \left(\mathbf{c}\mathrm{d}\mathbf{m}_t^\top + \mathrm{d}\mathbf{m}_t \mathbf{c}^\top\right)[\mathrm{diag}\,(\mathbf{c})\,\mathbf{S}\mathrm{diag}\,(\mathrm{d}\mathbf{m}_t) + \mathrm{diag}\,(\mathrm{d}\mathbf{m}_t)\,\mathbf{S}\mathrm{diag}\,(\mathbf{c})],$$

$$= \mathbf{c}\mathbf{s}^\top \mathbf{S}\mathrm{diag}\,(\mathbf{c})\,\mathrm{d}t - \mathbf{D}\mathrm{diag}\,(\mathbf{S}\mathrm{diag}\,(\mathbf{c})\,\mathbf{c})\,\mathrm{d}t + \mathbf{D}\mathrm{diag}\,(\mathbf{c})\,\mathbf{S}\mathrm{diag}\,(\mathbf{c})\,\mathrm{d}t.$$

$\square$

**Lemma D.5.** *With $\mathbf{S}$ defined in Equation* (D.1)*, $\mathrm{d}\mathbf{F} = \mathrm{d}\mathbf{F} = \boldsymbol{\Sigma}\mathbf{V}^\top \mathrm{d}\mathbf{B}_t \mathbf{c}^\top + \mathbf{c}\mathrm{d}\mathbf{B}_t^\top \mathbf{V}\boldsymbol{\Sigma}$ and $\mathrm{d}\mathbf{m}_t \stackrel{\mathrm{def}}{=} (\boldsymbol{\sigma} \odot \mathrm{d}\tilde{\mathbf{B}}_t)$.*

$$(\mathbf{S} \odot \mathrm{d}\mathbf{F})(\mathbf{S} \odot \mathrm{d}\mathbf{F}) = \mathbf{D}\mathrm{diag}\left(\mathbf{S}\mathrm{diag}\,(\mathbf{c})^2\,\mathbf{S}\right)\mathrm{d}t + \mathrm{diag}\,(\mathbf{c})\,\mathbf{S}\mathbf{D}\mathbf{S}\mathrm{diag}\,(\mathbf{c})\,\mathrm{d}t. \tag{D.7}$$

*Proof.*

$$(\mathbf{S} \odot \mathrm{d}\mathbf{F}) = \mathbf{S} \odot \left(\mathrm{d}\mathbf{m}_t \mathbf{c}^\top + \mathbf{c}\mathrm{d}\mathbf{m}_t^\top\right),$$

$$= \mathrm{diag}\,(\mathbf{c})\,\mathbf{S}\mathrm{diag}\,(\mathrm{d}\mathbf{m}_t) + \mathrm{diag}\,(\mathrm{d}\mathbf{m}_t)\,\mathbf{S}\mathrm{diag}\,(\mathbf{c}).$$

Now, computing the product,

$$(\mathbf{S} \odot \mathrm{d}\mathbf{F})(\mathbf{S} \odot \mathrm{d}\mathbf{F}) = [\mathrm{diag}\,(\mathbf{c})\,\mathbf{S}\mathrm{diag}\,(\mathrm{d}\mathbf{m}_t) + \mathrm{diag}\,(\mathrm{d}\mathbf{m}_t)\,\mathbf{S}\mathrm{diag}\,(\mathbf{c})][\mathrm{diag}\,(\mathbf{c})\,\mathbf{S}\mathrm{diag}\,(\mathrm{d}\mathbf{m}_t) + \mathrm{diag}\,(\mathrm{d}\mathbf{m}_t)\,\mathbf{S}\mathrm{diag}\,(\mathbf{c})],$$

$$= \mathbf{D}\mathrm{diag}\left(\mathbf{S}\mathrm{diag}\,(\mathbf{c})^2\,\mathbf{S}\right)\mathrm{d}t + \mathrm{diag}\,(\mathbf{c})\,\mathbf{S}\mathbf{D}\mathbf{S}\mathrm{diag}\,(\mathbf{c})\,\mathrm{d}t.$$

$\square$

**Lemma D.6.**

$$(S \odot \mathrm{d}\mathbf{F})\mathrm{d}\mathbf{m}_t \mathbf{c}^\top \mathrm{d}\mathbf{F} =$$

*Proof.*

$$(S \odot \mathrm{d}\mathbf{F})\mathrm{d}\mathbf{m}_t = [\mathrm{diag}\,(\mathbf{c})\,\mathbf{S}\mathrm{diag}\,(\mathrm{d}\mathbf{m}_t) + \mathrm{diag}\,(\mathrm{d}\mathbf{m}_t)\,\mathbf{S}\mathrm{diag}\,(\mathbf{c})]\,\mathrm{d}\mathbf{m}_t = \mathrm{diag}\,(\mathbf{c})\,\mathbf{S}(\boldsymbol{\sigma} \odot \boldsymbol{\sigma})$$

$\square$

