# OpenReview forum: "SGD vs GD: Rank Deficiency in Linear Networks"
_NeurIPS.cc/2024/Conference — NeurIPS 2024 poster_

### Official Review · Reviewer_kpcH · 2024-07-09

**Soundness:** 3
**Presentation:** 2
**Contribution:** 2
**Rating:** 6
**Confidence:** 3

**Summary:**

This paper presents an interesting dichotomy between SGD vs. GD, and how stochasticity in the gradients have the ability to implicitly regularize towards low-rank solutions. I believe this point is best highlighted by Theorem 5.1, where the show that the highlighted term in Equation (5.6) has a repulsive force, increasing the gap between larger and smaller eigenvalues. The "punchline" of the paper is technically given in Theorems 4.1 and 4.2, where they show that the solutions given by GD cannot have its rank diminished (rather stays at initialization), whereas we can see a decay in SGD (or SGF rather).

**Strengths:**

- I believe this paper fits well within the study of implicit regularization properties of certain optimizers and how they drive generalization. To this end, it also includes a related works section.
- The results are interesting, albeit a little limiting.
- I think the result in Theorem 4.2 also highlights how large step sizes can be beneficial for generalization. This is quite a popular topic in the literature. It is easy to see that when $\eta$ increases, the determinant vanishes faster.

**Weaknesses:**

- I think the presentation of the paper can be slightly improved. For example, the discussion on initialization on Page 4 was a bit confusing. The authors discuss different initialization tactics and the balancing effect, but they do not clarify which initialization they ultimately chose. I believe the intention was to show how initialization plays a role in Theorems 4.1 and 4.2, but this could be made clearer if that was indeed the intention. Furthermore, the authors could also change the notation to $\Delta(t)$ between lines 149-150. The authors are also missing a > in line 228.
- The experiments section is a bit lacking. Are there ways we can leverage the results from this paper to more practical scenarios / networks?
- My main concern (which is also the weakness) is the similarity between this paper and [1], which I don't believe the authors cited. From what I read, the main takeaway in [1] is that the noise in SGD has the ability to "push" or "attract" networks towards simpler networks, which they describe to be invariant sets. If you look at Sections 5 and 6 in [1], I believe what they are trying to say is that the noise has the ability to collapse the network to an invariant set that is low-rank, and one in which some singular modes are actually never learned. At a high-level, it seems to me that this might be a stronger result than to say that there exists a repulsive force between the eigenvalues, which is the current result. If I am missing something, please feel free to correct me. Though, at the very least, I think the authors should cite this paper and delineate the similarities / differences.

[1] Feng Chen, Daniel Kunin, Atsushi Yamamura, Surya Ganguli. "Stochastic Collapse: How Gradient Noise Attracts SGD Dynamics Towards Simpler Subnetworks". NeurIPS 2023.

**Questions:**

See Weaknesses. I would be more than happy to raise my score if my concerns can be addressed.

**Limitations:**

I have stated the limitations throughout my review.

---

> ### Author Rebuttal · Authors · 2024-08-07
>
> We thank the reviewer for their valuable comments, and for pointing out the missing reference.
>
> > The result in Theorem 4.2 also highlights how large step sizes can be beneficial for generalization.
>
> Yes, the presence of $\eta$ does control the rate of decrease of the determinant. Hence, the larger the step size, the greater the regularization effect, which can be beneficial for generalization. Note however that due to our choice of continuous-time modeling of gradient methods, as the step size increases, the discretization error makes it difficult to accurately capture the trajectories of the discrete methods with their continuous-time counterparts, leading to potentially different behavior. In the revision, we will add a statement indicating that our model captures the role of step size up to a certain order.
>
> > Initialization, balancedness and presentation.
>
> Thank you for the suggestion and the comments on the presentation of the initialization and its balancing effects. We address these points here and will revise the manuscript to incorporate these comments, making the presentation clearer.
>
> We mentioned initialization in the problem setup because it is an important ingredient in studying the implicit bias of homogeneous networks, including linear and ReLU networks. Previous works offer various initialization strategies for gradient flow, where certain structures emerge due to the balancedness properties of gradient flow.  In contrast,  with stochastic methods, the initialization does not matter. The emergence of low-rank structures is observed despite general initialization. This behavior is exemplified in Theorem 4.1 and 4.2, which hold for any initialization. In addition, we discuss these different initialization strategies to highlight the challenges for stochastic gradients, as the balanced quantities studied for gradient flow are insufficient.
>
> > Limited experimentation, more practical networks.
>
> We kindly refer the reviewer to the general comment regarding this limitation.
>
> > Detailed comparison with [1]
>
> Thank you for pointing out this work. We were aware of it but unfortunately missed including the reference and detailed comparison in the manuscript. We will revise it accordingly.
>
> [1] investigates a phenomenon similar to ours, referred to as stochastic collapse, where the noise in the gradients causes the iterates to collapse onto certain invariant sets.  [1] also uses continuous-time modelling of SGD as an SDE. The differences between the works are outlined as follows:
>
> - In the first part of their work, [1] provides a condition under which an invariant set is attractive for the SDE they consider. Methodologically, their work characterizes the **local behavior** around the invariant sets. Specifically, once the iterates enter a small
> $\epsilon$-neighborhood of these sets, they are attracted towards the sets. However, their does not address whether the iterates ever enter such an $\epsilon$-neighborhood for a general initialization.
>
>   In contrast, we offer a **global guarantee**, albeit within a simplified model: low-rank behavior is observed for any initialization. To establish these global properties, we developed a novel approach by tracking a specific quantity (the determinant).  This approach enables us to characterize the global behavior and reveal the dichotomy between the presence and absence of noise.
>
>
> - The paper also studies linear networks in a teacher-student setup.  Under a set of assumptions A1-A4~[1, p.30], they derive the evolution of singular values. However, due to the balanced spectral initialization (A3-A4) and structured label-noise (A2), the analysis falls short of capturing the repulsive force in the singular values. On a technical front, deriving the SDE for the singular values is simpler when the singular vectors remain stationary, as their assumptions ensured. When the singular vectors do not move, all the matrices can be simultaneously diagonalized, which simplifies deriving the singular values' dynamics. In our problem, however, the singular vectors move and follow a stochastic process, which makes tracking them significantly more challenging.
>
> [1] Feng Chen, Daniel Kunin, Atsushi Yamamura, Surya Ganguli. "Stochastic Collapse: How Gradient Noise Attracts SGD Dynamics Towards Simpler Subnetworks". NeurIPS 2023.

---

> > ### Comment · Reviewer_kpcH · 2024-08-09
> > **Reviewer Response**
> >
> > Thank you for the clarifications, particularly with the missing citation. I now see clearly the difference in contribution. I think the proof techniques used in the paper, especially in tracking the determinant, is useful for future work and will raise my score.

---

### Official Review · Reviewer_cAQT · 2024-07-12

**Soundness:** 2
**Presentation:** 3
**Contribution:** 3
**Rating:** 6
**Confidence:** 3

**Summary:**

This paper studies and analyzes how the rank of the parameter matrices evolves for two-layer linear networks when using GD and label noise SGD. The paper basically shows that while GD preserves the rank at initialization throughout its trajectory, SGD reduces the rank, thereby removing spurious directions. A stochastic differential equation (SDE) is derived to characterize the evolution of the eigenvalues of the parameter matrix to formalize the rank-reducing behavior of SGD. Small-scale experiments validate the derived theory.

**Strengths:**

The paper has a nice insight backed up by solid theory. I'm not an expert in this area but it seems that this is a novel insight.

**Weaknesses:**

**1.** While the rank-reducing property of SGD is interesting, it does not imply that the prediction error (which is the generalization metric most people would care about) of SGD is lower than that of GD. What if SGD converges to a completely erroneous low-rank solution? I understand that the paper is not directly about the generalization abilities of SGD and GD, but I feel this point should be discussed somewhere.

**2.** Since the results here are only for linear networks (which is completely fine), is there any intuition for why the results and claims of this paper should extend to non-linear networks?

**Questions:**

Please see the Weaknesses. Also, I have the following questions:

**1.** In line 174, is $M = \Theta^\top \Theta$?

**2.** In line 207 and the equation thereafter, if $W = U \Sigma V^\top$, then $W^\top a$ should be $V \Sigma U^\top a$.

**3.** I don't understand the jump from eq. (5.2) to (5.3). There seems to be an issue with the $\sqrt{\eta \delta}$ term (with the definition of $d X$).

**Limitations:**

Discussed somewhat throughout the paper.

---

> ### Author Rebuttal · Authors · 2024-08-07
>
> We would like to thank the reviewer for their insightful suggestions.
>
> > Low-rank solutions and prediction error.
>
> We agree with the reviewer's comment that the implicit regularization might only be useful for generalization if it is aligned with the ground truth model (i.e., in the classical example of sparse regression, $\ell_1$-regularization only improves generalization of ERM if the ground truth regressor is sparse). We also agree with the reviewer's comment regarding convergence to a sub-optimal low-rank solution which can happen due to possible over-regularization. We will discuss these issues in the conclusion of our manuscript.
>
>
> > Intuition for non-linear network.
>
> Empirically, the low-rank phenomenon is prominent for non-linear networks when trained with SGD, see Figure 2,3 in the attached pdf.
> For piecewise linear non-linearity like ReLU, each neuron's weights exhibit a multiplicative structure. Let $h (a,w, .) = \sum_{j} a_j \sigma(\langle{w_j}{.}\rangle) $ be the parametric model where $\sigma$ is the ReLU non-linearity and $(w_i,a_i)$ are the input and output weights of a neuron. With $\theta_i = \begin{bmatrix}
>     w_i^{\top} &  a_i
> \end{bmatrix}  $, the dynamics of the neuron can be rewritten as $\mathrm{d}{\theta_i} = \theta_i \mathrm{d}{J_i}$, for some curated matrix stochastic process $J_i$.
> For linear networks, these matrix-valued processes are identical across neurons, which makes the analysis tractable. In contrast, non-linear networks do not share this property. However, during the large noise phase, the stochastic matrices $J_i$ are dominated by noise even for the ReLU dynamics, which can result in some similarities between these matrices. This may vaguely explain the observed low-rank behavior.
>
>
> > Is $M = \theta^{\top}\theta$ in line 174 ?  The clarification of $W^{\top}a$ in line 207 ?
>
> Yes, thank you for pointing out this missing definition. We will correct it. In l.207, it should be $Wa$ instead of $W^{\top}a$.
>
> > The jump from eq. (5.2) to (5.3).
>
> It is possible due to changing the time, let $t' = (\eta \delta) t $. Now we have that $$ \mathrm{d}{t} = \frac{1}{\eta \delta} \mathrm{d}{t'}. $$ The Brownian motion has the following property $\mathrm{d}{\mathbf{B}_{t'}} = \sqrt{\eta \delta} ~ \mathrm{d} {\mathbf{B}}_t$  (the square root is due to different scaling of Brownian motion while changing time). Substituting these gives us the required  jump.

---

> > ### Comment · Reviewer_cAQT · 2024-08-10
> >
> > Thanks for the response!

---

### Official Review · Reviewer_Kp9L · 2024-07-12

**Soundness:** 4
**Presentation:** 4
**Contribution:** 4
**Rating:** 8
**Confidence:** 3

**Summary:**

This paper studies the implicit bias of SGD for two-layer linear networks. The authors study this primarily using (stochastic) gradient flow (S)GF, the continuous time version of (S)GD. To model the stochasticity, they approximate the SGD noise with independent label noise. Under these assumptions, they prove that the determinant of the Gram matrix of the parameters is invariant under GF, whereas it exponentially decays under SGF. Hence, if such a network is initialized with a full-rank parameter matrix, the rank asymptotically drops by at least one under label noise SGF, which lends some evidence to a low-rank bias for SGD not present in GD. The stochastic eigenvalue dynamics are investigated further, and while the exact dynamics are not rigorously pinned down, the paper calls out the repulsive forces that could explain why SGD converges to lower rank solutions. These results are empirically verified using some synthetic data experiments for regression and classification.

**Strengths:**

1. Laying theoretical foundations for the benefit of SGD over GD for optimizing neural networks is an important direction for the theory of deep learning. This paper takes a step towards identifying the implicit bias of SGD.
2. Previous work has that balancedness of adjacent layers is a conserved quantity for deep linear neural networks optimized using GF. Invariants of the dynamics are an important analytic tool; balancedness is crucial for proving convergence for deep linear networks. This paper identifies a very clean invariant for GF which separates GF from (label noise) SGF: the determinant of the Gram matrix of a certain concatenation of the linear layers ($\Theta$ in the paper). In particular, although the determinant is conserved for GF, for SGF the determinant decays exponentially (exactly!). This is a very nice contribution to the theory of SGD.
3. By itself, the determinant decay only establishes that the rank drops by one in SGF. However, the authors also investigate the behavior of the singular values of the first layer $W$ in a simpler scalar regression setting by deriving the SDE for the eigenvalues of $M = W^\top W$. Although they are not able to characterize the solution to this SDE, they clearly interpret the forces at play and highlight a repulsive effect between the eigenvalues of $M$, which encourage the singular values to decay towards zero.
4. The authors also consider the large noise limit and cleverly reduce to the 2-dimensional setting, where having a zero singular value is equivalent to the two rows being multiples of each other. They prove that, in expectation, (a power of) the larger singular value does not decrease, whereas the (same power of the) smaller singular value does not increase.
5. Section 6 contains many interesting extensions to other settings, such as classification, directly modeling the SGD noise as Gaussian with matching covariance, and discrete time methods.
6. The paper is quite transparent with the limitations of the theory. It also clearly highlights where the future directions are (e.g., the full characterization of the solution for the SDE governing the evolution of eigenvalues).

**Weaknesses:**

It was nice to see some experiments validating the theory, but I felt that the dimensionality of the problems should be increased to be more convincing. Right now the experiments are done with $(p, l, k) = (5, 10, 2)$. Even increasing these to being in the hundreds would be interesting, and correspondingly it might be appropriate to use other metrics to measure the rank of $W_1$, such as the effective rank  $\tr(M)/||M||$.

**Questions:**

1. In line 95, should $d$ and $p$ be swapped? Since the input is $p$-dimensional, whereas the parameter $\theta$ is $d$-dimensional (for example, the two-layer setting $d = p \cdot k$).
2. In line 96, there should be an $\ell_2$ norm outside of $y_i - f_\theta(x_i)$.
3. In line 108, $k=1$ should be in math mode.
4. There is a minor typo on the equation after line 208: it should read $Wa$ rather than $W^\top a$.
5. In the statement of Theorem 5.1 on Line 228, there is a missing comma in the definition of the ordered eigenvalues.
6. Could the authors add a reference to the proofs in the appendix when the result is stated in the main text (e.g. Lemma 6.1)?
7. In Section 7, I think I am missing something obvious: why are the $d + \ell - k$ smallest singular values equal $\sigma_2, \sigma_3, \sigma_4$, since to my understanding $d + \ell - k = 10 + 10 - 2 = 18$?
8. What can one hope to say about deeper linear networks? If I understand correctly, if one naively generalizes the construction of $\Theta$ then the special block structure of the label noise portion will no longer hold (in particular, it seems that $Theta$ would need to contain all the prefix and suffix products of the linear layers)

**Limitations:**

Yes

---

> ### Author Rebuttal · Authors · 2024-08-07
>
> We thank the reviewer for their valuable feedback, encouraging remarks, and meticulous proofreading of our work. They will help to make our work clearer.
>
> > It was nice to see some experiments validating the theory, but I felt that the dimensionality of the problems should be increased to be more convincing. Right now the experiments are done with $(p,l,k) = (5,10,2)$. Even increasing these to being in the hundreds would be interesting, and correspondingly it might be appropriate to use other metrics to measure the rank such as the effective rank.
>
> We refer the reviewer to the general comment for the experiments with higher dimensions.
>
> > In Section 7, I think I am missing something obvious: why are the smallest singular values equal
> $\sigma_2, \sigma_3, \sigma_4$. since to my understanding $d + l - k = 18$ ?
>
>  Apologies for the typo. As we are measuring the singular values of $W_1$, it should be the $p - k$ smallest singular values instead of the $p + l -k$ ones . Here $p=5$, $l = 10 (>p)$ and  $k=2$, hence the $p - k = 3$ smallest singular values are $\sigma_2,\sigma_3,\sigma_{4}$.
>
> > What can one hope to say about deeper linear networks? If I understand correctly, if one naively generalizes the construction of $\Theta$
>  then the special block structure of the label noise portion will no longer hold (in particular, it seems that $\Theta$
>  would need to contain all the prefix and suffix products of the linear layers)
>
> Thank you for this question. Indeed, deeper layers cannot be directly written in this multiplicative format. However, if the network has $l$ layers $W_1,\ldots,W_l$, for any adjacent layers $W_{i}$ and $W_{i+1}$, we can form a block matrix, $\theta_{i} = \begin{bmatrix} W_{i}^{\top} & W_{i+1} \end{bmatrix}$ and the multiplicative structure can be seen in the evolution of $\theta_i$. The analysis can then be performed with these block structures. For the stochastic case, however, the noise covariance is not straightforward and needs to be handled carefully.
> This approach might present a preliminary way forward for deeper networks. Another approach could be to carefully formulate a tensor structure that extends the spirit of our work beyond two layers.

---

> > ### Comment · Reviewer_Kp9L · 2024-08-07
> >
> > Thank you for the additional experiments and clarifications; I am definitely convinced by the plots you included in the supplemental figures.
> > Also, I appreciate the thoughtful response to my question about deeper linear layers, and would love to see any future work in this direction.
> > Cheers!

---

### Official Review · Reviewer_V8cQ · 2024-07-13

**Soundness:** 3
**Presentation:** 4
**Contribution:** 3
**Rating:** 7
**Confidence:** 3

**Summary:**

The paper proposes an analysis of the gradient flows of two layer linear neural networks with a squared loss borrowing tools from differential equations. The paper's result establishes that the stochastic gradient method generates solutions (limit of the flow as time goes to infinity) with determinant of the parameters x parameters transpose tending to zero. This implies that the parameters of a model optimized with stochastic gradient converge to a low-rank (simple) solution. While the full gradient method's solutions for the same problem do not exhibit the same simple structure. Authors conclude that this finding gives an explanation to the implicit bias of SGD.

**Strengths:**

The paper is tackling a very interesting problem and in a relatively simple yet representative setup where learnings can be used for deriving broader intuitions on the reasons behind SGD's implicit bias. The paper's methodology looks solid and the conclusions in the proposed context are well presented.

**Weaknesses:**

The paper studies a very simple model. Numerical experiments are also on the same and in very small data. It would benefit the paper to run experiments on larger / more complex models to discuss the limits and validity of this theory. The generalization to other settings section 6 is only changing the loss function. The section's title is perhaps overpromising.

**Questions:**

Is it possible to verify the breadth of validity of the result (parameters of SGD based solution live in a low-rank manifold, not the FGD) on different neural network architectures and see whether similar conclusions hold? Or which other property (if low rank is too restrictive) would you test for?

Does this theory imply that for the simpler vector problem the solution of SGD is a sparse vector where FGD finds dense solutions?


If we add a ReLU non-linearity between W1 and W2, then in what form the simplicity of the solution by SGD will get modified? locally low-rank?

**Limitations:**

The papers does not discuss how broadly their findings are valid. Numerical experiments could be used for testing whether findings are still valid in slightly more complex models

---

> ### Author Rebuttal · Authors · 2024-08-07
>
> We thank the reviewer for their interesting questions and encouraging comments.
>
>  > Numerical experiments are also on the same and in very small data. It would benefit the paper to run experiments on larger / more complex models to discuss the limits and validity of this theory.
>
> Please refer to the general comment.
>
>   > Is it possible to verify the breadth of validity of the result (parameters of SGD based solution live in a low-rank manifold, not the FGD) on different neural network architectures and see whether similar conclusions hold? Or which other property (if low rank is too restrictive) would you test for?
>
> The phenomenon that the parameters of solutions found by SGD live in a low-rank manifold is already empirically observed in [1] (note that the noise is more prominent when the step size is large, hence the effects of SGD are prominent with large step size). Given that measuring the rank is too computationally expensive, metrics that measure the similarity between the activations or weights of the neurons would serve as a good alternative. Such metrics are used by [1] and [2].
>
> [1] M. Andruischenko, AV Varre, L. Pillaud-Vivien, N. Flammarion. SGD with large step
> size learns sparse features, ICML 2023.
>
> [2] Feng Chen, Daniel Kunin, Atsushi Yamamura, Surya Ganguli. "Stochastic Collapse: How Gradient Noise Attracts SGD Dynamics Towards Simpler Subnetworks". NeurIPS 2023.
>
> > Does this theory imply that for the simpler vector problem the solution of SGD is a sparse vector where FGD finds dense solutions?
>
> No, the theory does not directly imply it.
> Consider the vector problem of diagonal networks where $u \odot v$ is the re-parameterization considered.
> Let $U = \textrm{diag}(u)$, and  $V = \textrm{diag}(v)$ be diagonal matrices with diagonal entries  $u$ and $v$, respectively.
> The block structure can be recovered with $\theta = [ U ~ ~ V ] $.
> However, the dimensions of this block matrix are $p \times 2p $, hence our theory does not have any implications as we need $l > d+k$. However, if a complete characterization can be obtained, then indeed a low rank of $\theta$ would imply a sparse solution. The sparse vector problem with label noise has received much attention [3,4] and the complete characterization in the symmetric case (u=v) is given by [5].
>
>
>
> [3]J. Z. HaoChen, C. Wei, J. Lee, and T. Ma. Shape matters: Understanding the implicit bias of the noise covariance. COLT, 2021.
>
> [4] Z. Li, T. Wang, and S. Arora. What happens after sgd reaches zero loss? –a mathematical framework. ICLR 2022.
>
> [5] Pillaud-Vivien, L., Reygner, J., and Flammarion, N. Label noise (stochastic) gradient descent implicitly solves the lasso for quadratic parametrisation. COLT, 2022.
>
> > If we add a ReLU non-linearity between W1 and W2, then in what form the simplicity of the solution by SGD will get modified? locally low-rank?
>
> Please refer to the general comment, particularly the paragraph on ReLU networks, as the experiments indicate the simplicity bias that SGD tends to induce.

---

### Author Rebuttal · Authors · 2024-08-07

## Broader empirical evaluation
We would like to thank the reviewers for their positive assessment of the paper and their appreciation of our work. Below, we address general comments that were raised multiple times. Individual responses can be found following each review. All referenced figures can be found in the attached PDF.

**Higher dimensions.**
First, at the request of reviewer Kp9L, we have conducted experiments in higher dimensions to robustly evaluate our theoretical findings. We choose a scalar linear regression problem on Gaussian data in dimension $p = 100$. We trained a linear network with an inner layer width of $l = 100$ using gradient descent, both with and without label noise. As shown in Figure 1, we observe that when trained with label noise, all singular values of the $W_1$ matrix, except the largest one, diminish to zero. This behavior is in contrast to the deterministic full-batch gradient descent where the singular values do not diminish.

**Non-linear networks.**
Many reviewers have asked for broader empirical validation of the phenomenon we theoretically capture in linear networks.
Indeed, recent efforts have been made to empirically understand the regularization effects induced by stochastic noise across various architectures.
For the sake of completeness, we recall some empirical results here. First, we consider simple non-linear architectures and then discuss general architectures. This discussion will be added to the revised version of the manuscript.

**ReLU non-linearity.** (more relevant to the question of reviewers V8cQ, cAQT).
Following the approach of [1], we consider a one hidden-layer ReLU network in a teacher-student setup for a scalar regression task.
We first present empirical evidence for $p=2$, as this allows for clearer visualizations.
Figure 2 shows how training with label noise regularizes ReLU networks by pushing the neurons to align with a few relevant directions (the directions of the teacher neurons here).
As reviewer V8cQ pointed out, the weight matrix gets locally low rank and the neurons are aligned.
Once they are aligned, they stay aligned, thus inducing alignment globally.
In Figure 3, we show the singular values for a regression problem in dimension $p=5$.  The dynamics of the singular values are also similar to the case of linear networks we have theoretically studied.

**General architectures.**
For large-scale neural networks, we refer to the experiments in papers [1] and [2].
[1] studies implicit regularization across various architectures, from single hidden-layer networks to deep networks like DenseNet, while [2]  focuses on various deep learning architectures like ResNet and VGG.
The main takeaway from these works is the empirical verification of the low-rank phenomenon, which we aim to support with theoretical backing.
The low-rank phenomenon is verified by the sparsity coefficient in [1] and the fraction of independent neurons in [2], which measures the alignment between the weights or activations of the neurons.
Intuitively, the more aligned they are, the lower the rank of the parameters.
We will appropriately cite these empirical works throughout our paper to broaden the scope of our results.


[1] M. Andruischenko, AV Varre, L. Pillaud-Vivien, N. Flammarion. SGD with large step
size learns sparse features, ICML 2023.

[2] F. Chen, D. Kunin, A. Yamamura, S. Ganguli. "Stochastic Collapse: How Gradient Noise Attracts SGD Dynamics Towards Simpler Subnetworks". NeurIPS 2023.

---

### Decision · Program_Chairs · 2024-09-25

**Decision:**

Accept (poster)

**Comment:**

The paper investigates the differences between Stochastic Gradient Descent (SGD) and Gradient Descent (GD) in the context of a two-layer linear network with square loss. It reveals a dichotomy: GD maintains the rank of the parameter matrix from initialization, whereas SGD, especially in the presence of label noise, reduces the rank over time. This phenomenon is analyzed by examining the evolution of the determinant of the parameter matrix, and the stochastic differential equation (SDE) governing the eigenvalues is derived. The SDE indicates a repulsive force between eigenvalues, leading to rank deficiency. The findings are supported by experiments extending beyond linear networks and regression tasks.

The referees highlighted that the analysis was comprehensive: the derivation of the SDE governing eigenvalues adds depth to the understanding of rank deficiency, providing a strong mathematical foundation for the observed phenomena. I recommend to accept it to NeurIPS 2025.

Nevertheless, I believe the authors should take into account the issues raised by the referees for the camera-ready version, but it seems the authors acknowledged them during the rebuttal.